# DLEBench: Evaluating Small-scale Object Editing Ability for Instruction-based Image Editing Model

**Shibo Hong** [* 1]  **Boxian Ai** [* 1]  **Jun Kuang** [2]  **Wei Wang** [2]  **FengJiao Chen** [2]  **Zhongyuan Peng** [1]  **Chenhao Huang** [1]  **Yixin Cao** [1]

## Abstract

Significant progress has been made in the field of Instruction-based Image Editing Models (IIEMs). However, while these models demonstrate plausible adherence to instructions and strong reasoning ability on current benchmarks, their ability to edit small objects remains underexplored, despite its importance for precise local editing and refining details in both real and generated images. In this paper, we introduce **D**eep**L**ook**E**dit**Bench** (DLEBench), the first benchmark dedicated to assessing the abilities of IIEMs in editing small-scale objects. Specifically, we construct a challenging testbed comprising 1889 samples across seven instruction types. In these samples, target objects occupy only 1%-10% of the image area, covering complex scenarios such as partial occlusion and multi-object editing. To ensure robust evaluation on this benchmark, we propose an evaluation protocol with refined score rubrics to minimize subjectivity and ambiguity in two criteria: Instruction Following and Visual Consistency. This protocol also introduces a dual-mode evaluation framework (Tool-driven and Oracle-guided Modes) addressing the misalignment between LMM-as-a-Judge and human judgments on DLEBench. Empirical results on 10 IIEMs reveal significant performance gaps in small-scale object editing, highlighting the need for specialized benchmarks to advance this ability.

## 1. Introduction

Instruction-based Image Editing Models (IIEMs) (Fu et al., 2023; Chen et al., 2025; Ji et al., 2025; Yu et al., 2025a; Geng

---
[*]Equal contribution [1]College of Computer Science and Artificial Intelligence, Fudan University [2]Meituan Group. Correspondence to: Yixin Cao <yxcao@fudan.edu.cn>.

*Proceedings of the 43rd International Conference on Machine Learning*, Seoul, South Korea. PMLR 306, 2026. Copyright 2026 by the author(s).

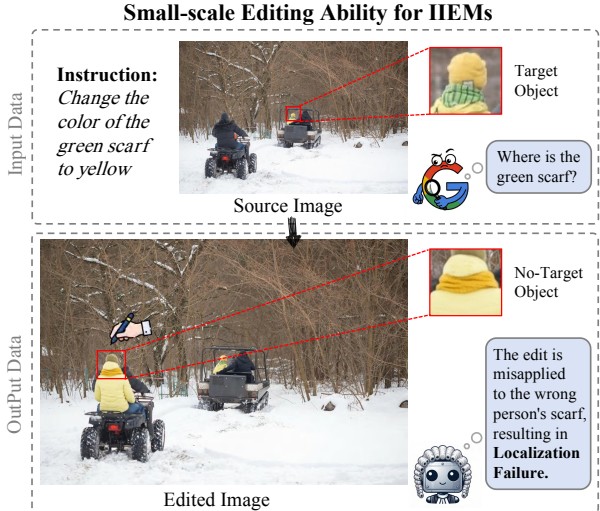

**Small-scale Editing Ability for IIEMs**

*Figure 1.* An example illustrating the challenge of small-scale editing. Given the instruction to edit the green scarf, Gemini-3-Pro misidentifies the target and modifies the foreground object instead.

et al., 2024; Yu et al., 2025b) offer superior usability over Mask-based Image Editing Models (MIEMs) (Couairon et al., 2022; Gu et al., 2022; Wang et al., 2023) by requiring only text instructions instead of additional masks. This user-friendly nature has driven significant progress in IIEMs.

The rapid advancement of IIEMs has driven the parallel development of benchmarks to comprehensively evaluate their abilities. Early benchmarks (Ma et al., 2024) focus on single-turn, single-object editing, evolving into multi-turn, multi-object settings (Ye et al., 2025) to test instruction adherence. Recent benchmarks have expanded further to evaluate reasoning abilities (Zhao et al., 2025), including temporal and logical deduction. However, despite the increasing complexity of editing and the heightened reasoning demands of current benchmarks, they exhibit a pervasive bias towards editing salient objects with dominant spatial footprints. We argue that as the object's spatial extent decreases, the editing paradigm shifts from broad modification to localized editing, posing a substantial challenge for IIEMs in small-scale object editing. As illustrated in Figure 1, even state-of-the-art IIEMs like Gemini-3-Pro fail to accurately

localize the target object, resulting in unsuccessful edits. Significantly, addressing this challenge is vital for generation tasks. It enables the targeted correction of small-scale errors, avoiding the computational waste and inconsistency of regenerating the image from scratch.

To evaluate the small-scale object editing ability of IIEMs, we aim to establish a dedicated benchmark. However, there are two challenges: First, while images containing small-scale objects exist in visually dense reasoning datasets, paired editing instructions tailored to them are scarce. Manually curating these pairs is labor-intensive. Second, evaluating small-scale object editing remains challenging. Conventional similarity metrics (e.g., CLIP) poorly align with human judgment on fine-grained details. Furthermore, even advanced LMM-based evaluators (i.e., LMM-as-a-Judge (Zheng et al., 2023; Hong et al., 2026)) fail to discern minute visual changes and thus yield unreliable assessments.

To address the first challenge, we design a semi-automated three-stage transformation pipeline to convert visual reasoning samples into image editing samples. First, we employ a counterfactual synthesis strategy to convert QA pairs into metadata, including editing instructions, instruction types, target objects, and image captions. Second, to address the challenge of generating reliable ground-truth reference images for small object editing, we introduce a crop-and-edit mechanism. This strategy isolates the target region for precise manipulation, enabling the generation of high-quality reference images that remain elusive for SOTA IIEMs operating on full images. Finally, the pipeline culminates in human verification to ensure data quality. Based on this pipeline, we construct DeepLookEditBench (DLEBench), the first benchmark dedicated to evaluating the small-scale object editing ability of IIEMs. It comprises 1,889 samples with target object areas limited to the 1%–10% range and incorporating complex scenarios, such as partial occlusion and multi-object editing, across seven instruction types.

To address the second challenge, we propose an evaluation protocol tailored for small-scale object editing. First, we employ two criteria: Instruction Following (IF), which evaluates the ability of IIEMs to edit small-scale objects, and Visual Consistency (VC), which assesses the preservation of global non-target regions. Crucially, existing score rubrics for IF and VC suffer from subjectivity and vagueness (e.g., relying on ambiguous quantifiers like 'most'), leading to unreliable assessments. To overcome this, we establish objective score rubrics defined by distinct failure modes (e.g., localization failure, wrong action). This redesign improves the reproducibility of the evaluation results and enables precise error diagnosis. Second, for evaluation methods, we propose a dual-mode evaluation framework to address the low alignment of the LMM-as-a-Judge with human judgments on DLEBench. Specifically, the Tool-driven Mode

ensures practicality by invoking external tools for iterative assessment, enabling evaluation without human intervention. Conversely, the Oracle-guided Mode prioritizes reliability by leveraging human-annotated bbox to isolate target objects, thereby eliminating localization errors. Experimental results demonstrate that our evaluation methods achieve greater alignment with human judgments than LMM-as-a-Judge, providing a reliable automated assessment.

In summary, our main contributions are as follows: (1) We propose DLEBench, the first benchmark dedicated to evaluating the small-scale object editing abilities of IIEMs, establishing a foundation in this field. (2) We design an evaluation protocol incorporating refined score rubrics and a dual-mode framework to ensure reliable assessment for small-scale editing. (3) We conduct a comprehensive evaluation and analysis of 10 representative IIEMs, offering novel insights into their small object editing ability.

**Conflict of Interest Disclosure.** The authors Jun Kuang, Wei Wang, and Fengjiao Chen are employed by Meituan Group. We clarify that this research did not involve the evaluation of any proprietary models or technologies developed by Meituan Group. This study was conducted independently, and the corporate affiliation of these authors did not influence the experimental design, data analysis, or the conclusions presented herein. All other authors declare no financial or substantive conflicts of interest.

## 2. Related Work

### 2.1. Benchmarks for Image Editing

To evaluate the evolving capabilities of IIEMs, benchmarks have progressed from basic single-turn editing to complex, reasoning-intensive scenarios. Benchmarks such as I2EBench (Ma et al., 2024) establish the foundational baseline for evaluating standard single-turn, single-object editing capabilities. As the field advanced, the focus shifted toward more complex and realistic settings: PIE-Bench++ (Huang et al., 2024) and GIE-Bench (Qian et al., 2025) target multi-object and grounded editing, while ImgEdit (Ye et al., 2025) and ChatEdit (Cui et al., 2023) introduce multi-turn interactive protocols to mimic iterative user workflows. Most recently, benchmarks have expanded to higher-order reasoning-intensive tasks, with suites such as KRIS-Bench (Wu et al., 2025c) and UniREditBench (Han et al., 2025) assessing adherence to physical knowledge and logical constraints. However, while these benchmarks emphasize semantic and logical complexity, they generally overlook the fine-grained visual perception and precise editing abilities required for small-scale objects. Such scenarios are particularly challenging because even slight localization errors or subtle distortions can lead to noticeable failures in editing quality and fidelity.

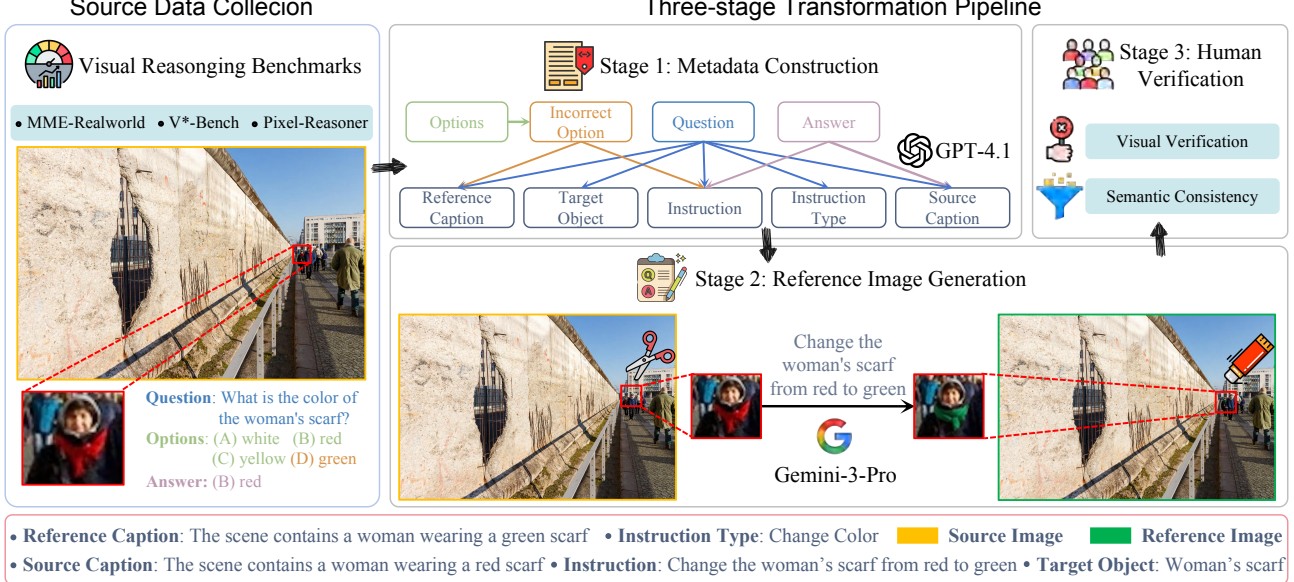

*Figure 2.* Overview of the three-stage data transformation pipeline. We begin by selecting a raw visual reasoning sample from V\*-Bench, specifically one that inquires about the color of a woman's scarf. In Stage 1, we employ a counterfactual synthesis strategy via GPT-4.1 to generate the image-edit metadata (as shown in the bottom red box). Subsequently, Stage 2 utilizes a crop-and-edit strategy with Gemini-3-Pro to generate the reference image. Finally, all data is submitted to Stage 3 for rigorous human verification.

## 2.2. Image Editing Evaluation

Early evaluation protocol (Zhang et al., 2023; Sheynin et al., 2024) primarily relied on similarity-based metrics (e.g., CLIP scores), which often correlate poorly with human judgments in complex editing scenarios. Consequently, recent benchmarks have adopted the LMM-as-a-judge paradigm (Ye et al., 2025; Han et al., 2025; Ma et al., 2024) that utilize strong Large Multimodel Models (LMMs) to assess results across three core dimensions: Instruction Following, Visual Consistency, and Visual Quality. However, these criteria frequently suffer from vague definitions and scoring rubrics, leading to significant subjectivity and ambiguity in the evaluation process. Furthermore, even advanced LMMs struggle with the fine-grained visual perception required by our benchmark. When editing targets occupy only a minute fraction of the image, standard LMM evaluators often fail to distinguish subtle modifications from background noise, necessitating a more rigorous and resolution-aware evaluation protocol. This challenge is further amplified in cluttered scenes or low-salience regions, where reliable assessment depends on accurately capturing both local visual changes and their consistency with the surrounding context.

## 3. DLEBench

In this section, we introduce DLEBench, a benchmark for evaluating the small-scale object editing ability of IIEMs. We detail the benchmark construction in Sec. 3.1, followed

by the benchmark statistics in Sec. 3.2.

## 3.1. Benchmark Construction

To construct DLEBench, we first require images where objects occupy only a small fraction of the scene. To this end, we collect images from visually dense reasoning benchmarks where the reasoning questions specifically target single or multiple small-scale objects embedded in cluttered environments. Furthermore, some of these objects are partially occluded, thereby serving as a rigorous testbed for accurately perceiving and subsequently editing them. We select three benchmarks as our primary sources: MME-Realworld (Zhang et al., 2024), Pixel-Reasoner (Wang et al., 2025), and V\*-Bench (Wu & Xie, 2023), aggregating to 2,043 raw visual reasoning samples. Each raw sample consists of an image $I$, a visual question $Q$, and a ground-truth answer $A$ (along with candidate options $O$), which serve as the basis for our subsequent data generation.

As illustrated in Figure 2, we devise a three-stage pipeline, comprising metadata construction, reference image generation, and human verification, to transform raw visual reasoning samples into structured image editing instances, denoted as $(S_I, S_C, T_O, Type, Instruction, R_I, R_C)$. Specifically, $S_I$ represents the source image containing the target object $T_O$, accompanied by its caption $S_C$. The editing task is defined by the *Instruction* and its specific category *Type*. Furthermore, $R_I$ denotes the ground-truth edited image (with caption $R_C$) serving as the evaluation reference

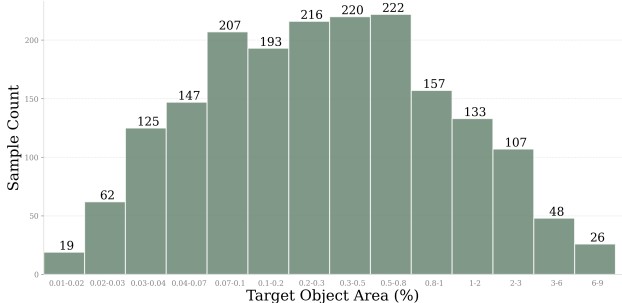

*Figure 3.* Comparison of the Cumulative Distribution Function (CDF) of target object area ratios across different instruction-based image editing benchmarks.

*Figure 4.* Distribution of target object area ratios in DLEBench. The bars represent the number of samples in different area intervals, with specific counts annotated above each bar.

against which the model-generated image $E_I$ is assessed.

**Metadata Construction.** In this stage, we employ a counterfactual synthesis strategy to transform the visual reasoning tuple $(Q, O, A)$ into the editing metadata $(S_C, T_O, Type, Instruction, R_C)$, as exemplified in Figure 2. Specifically, we first select an incorrect option $A_{neg}$ from $O$ to serve as the target attribute for the transition. By contrasting $A$ with this counterfactual state $A_{neg}$ within the context of $Q$, we synthesize the corresponding editing *Instruction*. Simultaneously, we generate $S_C$ and $R_C$ by converting the valid pair $(Q, A)$ and the counterfactual pair $(Q, A_{neg})$ into declarative statements, respectively, while directly extracting $T_O$ from the subject of $Q$. Finally, we categorize the editing *Type* based on the visual attribute queried in $Q$. This process yields a taxonomy of seven distinct editing instruction types, structured into attribute-level modifications (material, color, shape, OCR) and object-level modifications (count, replacement, removal). To implement this transformation at scale, we use GPT-4.1 guided by a crafted prompt (detailed in Table 15), leveraging its advanced capabilities in logical inference and natural language processing.

**Reference Image Generation.** The primary objective of this stage is to generate a high-quality reference $R_I$ for evaluation purposes, simultaneously verifying the validity of the generated *Instruction*. Initially, we treated the raw $I$ as the source $S_I$ and attempted the standard practice of applying the *Instruction* directly to $S_I$ using IIEMs. However, preliminary experiments revealed that even state-of-the-art IIMEs (e.g., Gemini-3-Pro) frequently fail to localize small-scale $T_O$ in a full-image editing setting.

To mitigate this, we adopt a crop-and-edit strategy that focuses the model's attention on the target region. However, this approach encounters two critical hurdles: (1) Inaccurate Localization: Even stronger detectors like GroundingDINO (Ren et al., 2024) or YOLOv13 (Lei et al., 2025) struggle to localize small-scale objects precisely; (2) Contextual Deficiency: For small-scale objects, tight crops deprive the model of global context, leading to boundary artifacts

and structural inconsistencies during reintegration.

We address these challenges through a dual approach. First, we rely on manual annotation to ensure precise bboxes for all targets. Second, we propose an adaptive bbox expansion strategy to balance local focus with global context. This strategy dynamically adjusts the crop size based on the object's size: incorporating broader context for smaller objects to facilitate seamless reintegration, while constraining expansion for larger objects to maintain localized focus. The expansion ratio $\lambda(s)$ is formalized as:

$$\lambda(s) = \begin{cases} \lambda_{\max} & \text{if } s \leq S_{\min}, \\ \lambda_{\min} & \text{if } s \geq S_{\max}, \\ (1 - \alpha)\,\lambda_{\max} + \alpha\,\lambda_{\min} & \text{otherwise}, \end{cases} \quad (1)$$

where $s = \min(w, h)$ represents the minimum dimension of the annotated bbox. We interpolate $\lambda(s)$ between a maximum ratio $\lambda_{\max}$ (for size $S_{\min}$) and a minimum ratio $\lambda_{\min}$ (for size $S_{\max}$) using a linear factor $\alpha = (s - S_{\min})/(S_{\max} - S_{\min})$. In practice, we set $\lambda_{\max} = 6.0$, $\lambda_{\min} = 0.3$, $S_{\min} = 32$, and $S_{\max} = 256$.

**Human Verification.** Upon completing the above stages, we implement a human verification stage to ensure the sample's quality. This procedure involves a two-fold inspection: (1) Semantic Consistency: We confirm that the human-annotated bbox accurately encloses $T_O$. Furthermore, they ensure that the metadata generated by GPT-4.1, specifically $(S_C, T_O, Type, Instruction, R_C)$, maintains strict semantic alignment with the original logic of $(Q, O, A)$. Any identified discrepancies are manually corrected. (2) Visual Verification: We verify whether the generated reference $R_I$ strictly adheres to *Instruction*, evaluating it against the criteria defined in Sec. 4.1. In cases of non-compliance, we regenerate $R_I$ up to three times. If the output remains unsatisfactory after these retries, we deem the editing instruction visually infeasible and discard the sample.

## 3.2. Benchmark Statistics

By employing our data transformation pipeline, we construct DLEBench, which comprises 1,889 small-scale object editing samples spanning seven distinct editing instruction types, as detailed in Appendix D. To highlight the unique focus of our benchmark on small-scale object editing, we compare the scale distribution of target objects against mainstream image editing benchmarks, including ImageEdit (Ye et al., 2025), UniREditBench (Han et al., 2025), RISE (Zhao et al., 2025), and KRIS-Bench (Wu et al., 2025c) (see Appendix E for more information). The cumulative distribution function (CDF) in Figure 3 shows that DLEBench is distinctly skewed towards smaller ratios, with the majority of target objects occupying less than 1% of the image area, while other benchmarks predominantly feature large-scale targets. This distinction underscores the unique value of our benchmark in evaluating small-scale editing capabilities.

Regarding the number of target objects per image, DLEBench predominantly comprises single-object edits (1,771 instances), complemented by a challenging subset of multi-object edits (118 instances) that involve up to 8 targets. Beyond target object counts, we analyze the distribution of samples by the relative area of the target objects. As illustrated in Figure 4, the sample distribution across the object area intervals approximates a log-normal distribution. Rather than being randomly scattered or heavily skewed towards extreme outliers, the distribution naturally centers around the $[0.07\%, 0.8\%)$ interval. Furthermore, the distribution smoothly tapers off towards the tails, spanning a broad spectrum from fine-grained modifications ($< 0.07\%$) to relatively large edits ($> 0.8\%$). This balanced structure ensures a robust, comprehensive evaluation across varying levels of spatial granularity.

## 4. Evaluation Protocol

### 4.1. Evaluation Criteria

To assess the small-scale editing ability of IIEMs, we use the **Instruction Following** (IF) criterion. This criterion assesses whether the model accurately localizes the target object and correctly executes the edit while strictly preserving intrinsic attributes irrelevant to the instruction. However, existing scoring rubrics for IF (Zhao et al., 2025; Wu et al., 2025c) are often limited by subjectivity and vagueness (e.g., "Score 2: Most of the required changes are missing"). These rubrics lead to unreliable assessments. To address these limitations, we introduce objective score rubrics defined by distinct failure modes. Unlike holistic ratings, our rubric enforces a hierarchical inspection process that ranks performance based on failure severity, ranging from Score 1 (Localization Failure), Score 2 (Wrong Action), Score 3 (Over Modification), to Score 4 (Flawless Execution). By strictly

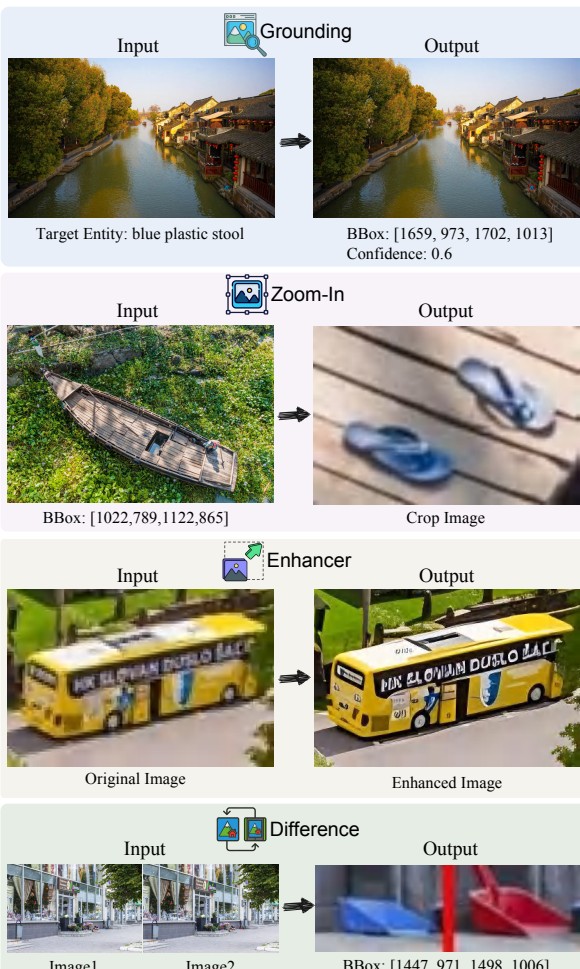

*Figure 5.* Overview of the input and output data for the tools employed in our Tool-driven Mode. These tools are also utilized when evaluating VC in the Oracle-guided Mode.

isolating localization errors from manipulation and preservation issues, this hierarchy ensures reproducible evaluation results and enables precise diagnosis of model bottlenecks.

In addition to IF, we incorporate **Visual Consistency** (VC), a critical criterion for image editing tasks. This criterion evaluates the integrity of non-target regions. It ensures that the non-target elements remain consistent. Mirroring the IF rubric, we redesigned the VC rubric as a hierarchical 4-point scale based on the severity of visual anomalies, ranging from Score 1 (Scene Collapse), Score 2 (Multiple Anomalies), Score 3 (Single Anomaly), to Score 4 (Perfect Consistency).

### 4.2. Dual-Mode Evaluation Framework

Our benchmark challenges not only IIEMs but also existing evaluation methods. Preliminary experiments (as illustrated in Figure 8) reveal that even advanced LMM-as-a-Judge methods (e.g., GPT-4.1 or Gemini-3-Pro) struggle to achieve high alignment with human judgments, stemming primarily

*Table 1.* Performance of different IIEMs across various instruction types on DLEBench. We use the Oracle-guided Mode to evaluate each output on two criteria: IF and VC. The final scores are then obtained by averaging the results of these two criteria (see Table 4 for the specific scores of each criterion). The best performance is highlighted in **bold**, and the second best is underlined.

| Instruction Types | Closed-Source Models | | Open-Source Models | | | | | | | |
|---|---|---|---|---|---|---|---|---|---|---|
| | Gemini-3-Pro | GPT-Image-1 | OmniGen2 | Bagel-Think | UniREdit-Bagel | MagicBrush | Qwen-Edit | UniWorld-V1 | UniWorld-V2 | Step1X-Edit |
| Change Material | **74.01** | 44.31 | 47.03 | 71.03 | 41.66 | 24.24 | 48.91 | 40.37 | 57.74 | 62.50 |
| Change Color | **66.99** | 37.87 | 33.58 | 62.24 | 39.47 | 24.32 | 44.52 | 30.69 | 55.21 | 57.93 |
| Change OCR | **66.54** | 37.28 | 38.93 | 49.46 | 29.69 | 18.39 | 36.79 | 28.65 | 47.79 | 59.39 |
| Change Shape | **68.84** | 46.38 | 41.31 | 67.75 | 43.02 | 28.99 | 33.34 | 35.51 | 53.63 | 60.87 |
| Removal Object | **73.76** | 50.52 | 52.67 | 67.29 | 48.65 | 28.41 | 55.72 | 35.50 | 52.53 | 63.90 |
| Replace Object | **64.86** | 40.37 | 48.23 | 63.36 | 37.62 | 29.43 | 41.13 | 35.00 | 57.58 | 57.45 |
| Change Count | 43.87 | 25.00 | 32.64 | **45.84** | 23.68 | 8.34 | 31.81 | 21.22 | 24.14 | 25.70 |
| Average | **65.55** | 40.25 | 42.06 | 61.00 | 37.68 | 23.16 | 41.75 | 32.42 | 49.80 | 55.39 |

from their weak visual perception. To address this limitation, we introduce a dual-mode evaluation framework, comprising two distinct configurations: (1) **Tool-driven Mode**: In this mode, the LMM acts as an agent to invoke external visual tools, actively compensating for its own perceptual limitations. (2) **Oracle-guided Mode**: the LMM operates on pre-processed data, utilizing human-annotated bbox and reference images to bypass localization errors. This dual-strategy effectively reconciles the trade-off between practicality and reliability. In the following sections, we detail the implementation of these two modes.

**Tool-driven Mode.** To mitigate the deficiency of LMMs in visual perception, we introduce a suite of tools (Shown in Figure 5) designed to bolster the LMMs' perceptual ability, including Grounding, Zoom-In, Difference, and Enhancer. Each tool generates either a transformed image or textual information, thereby providing more explicit visual or textual evidence to alleviate the model's perceptual deficiencies.

Specifically, we implement the Grounding tool using GroundingDINO (Ren et al., 2024) for object localization. However, since GroundingDINO often struggles to detect small-scale objects, we introduce a complementary Zoom-In tool to enable active spatial search. In this process, the LMM operates in a feedback loop: it iteratively generates and adjusts the bbox to invoke the Zoom-In tool, repeating the search until the target object is successfully located. In addition to the aforementioned localization-assisted tools, we introduce the Difference tool to find differences in two images. It compares the pixels of the two images to identify the locations of the variations, and outputs side-by-side cropped patches that emphasize these different regions. Finally, to ensure the visual clarity of all cropped images produced by the Zoom-In and Difference tools, we use the Enhancer tool, powered by Real-ESRGAN (Wang et al., 2021), which upsamples these crops to produce clearer inputs for evaluation.

For the evaluation pipeline of the Tool-driven Mode. Given $(S_I, Instruction, E_I)$, the LMM operates iteratively. At

each step $i$, the LMM generates a thought $T_i$ followed by an action $A_i$ or a final evaluation result $R$. Specifically, $A_i$ can be a JSON-formatted tool invocation command, and $R$ is based on aspect-specific score rubrics (See Sec. 4.1). In the case of a tool invocation, the execution yields an observation $O_i$, which is appended to the interaction history as input for the subsequent step. The process ends when the LMM outputs $R$ or reaches the maximum number of interaction turns. Formally, we define the LMM's inference process at step $i$ as follows:

$$T_i, \{A_i|R\} \sim E(\cdot \mid S_I, Instruction, E_I, \mathcal{H}_{i-1}), \quad (2)$$

where $\mathcal{H}_{i-1} = \{(T_j, A_j, O_j)\}_{j=1}^{i-1}$ denotes the interaction history up to step $i-1$, while $E$ signifies the LMM.

**Oracle-guided Mode.** Beyond relying on tools, we introduce an alternative evaluation mode that leverages human-annotated target object bboxes to preprocess the input images $(S_I, E_I, R_I)$, thereby decoupling evaluation from localization. Specifically, for IF, we crop $(S_I, E_I, R_I)$ around the target object $T_O$ to force the LMMs to focus exclusively on the edited region. Conversely, for VC, we mask $T_O$ with white pixels while providing the same Tool set as in Tool-driven Mode to assist evaluation. This strategy eliminates the target's visual influence, allowing the LMM to assess the consistency of non-target regions without distraction.

# 5. Experiments

## 5.1. Experimental Settings

To comprehensively assess the small-scale object editing ability of IIEMs, we curate a diverse set of representative models covering distinct architectures. We categorize them into three groups: (1) Auto-regressive Models: **Omni-Gen2** (Wu et al., 2025b), **Bagel-Think** (Deng et al., 2025), and **UniREdit-Bagel** (Han et al., 2025); (2) LMM-based Diffusion Models: **MagicBrush** (Zhang et al., 2023), **Qwen-Edit** (Wu et al., 2025a), **UniWorld-V1** (Lin et al., 2025), and **UniWorld-V2** (Li et al., 2025), which employ an LMM as the text encoder coupled with a diffusion transformer

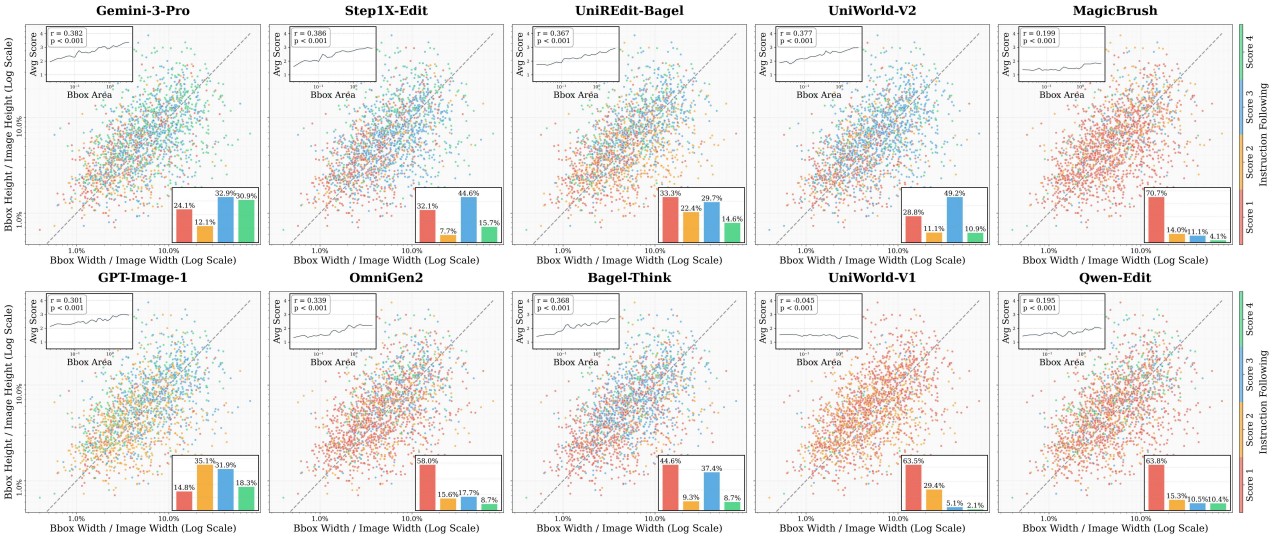

*Figure 6.* Impact of object scale on performance. The central scatter plots illustrate the relationship between the target object's relative scale (log scale) and IF score. The top-left inset displays the trend of average scores as the bbox area increases, along with the Pearson correlation coefficient ($r$) and statistical significance ($p$). The bottom-right inset presents the overall distribution of scores.

backbone; and (3) Hybrid Architectures: represented by **Step1X-Edit** (Liu et al., 2025), combining LMM capabilities with a DiT-style diffusion framework. Additionally, we include two proprietary models, **GPT-Image-1** and **Gemini-3-Pro**, accessed via their official APIs. To ensure evaluation accuracy, we employ the Oracle-guided Mode to assess all models across both the IF and VC. Furthermore, we normalize all scores to a 100-point scale to facilitate direct comparison across models.

### 5.2. Benchmarking Results on DLEBench

**Overall Performance.** Table 1 presents the quantitative results of different IIEMs on DLEBench across instructions. These results challenge the prevailing assumption that closed-source models consistently outperform their open-source counterparts. While Gemini-3-Pro secures the top rank with an average score of 65.55, demonstrating robust overall capability, select open-source models exhibit remarkable competitiveness. Notably, the open-source Bagel-Think (61.00) surpasses the proprietary GPT-Image-1 (40.25) by a substantial margin. This suggests that in specific small-scale editing contexts, proprietary models do not necessarily maintain absolute dominance, unlike in general reasoning benchmarks (Wu et al., 2025c; Han et al., 2025).

In terms of performance across different instructions, we observe consistent degradation in Change Count scores across all models. We attribute this to the inherent complexity of accurately isolating and enumerating multiple small target objects. Unlike instructions that edit a single small-scale instance, the Change Count requires a higher level of visual perception to handle multiple small entities simultaneously,

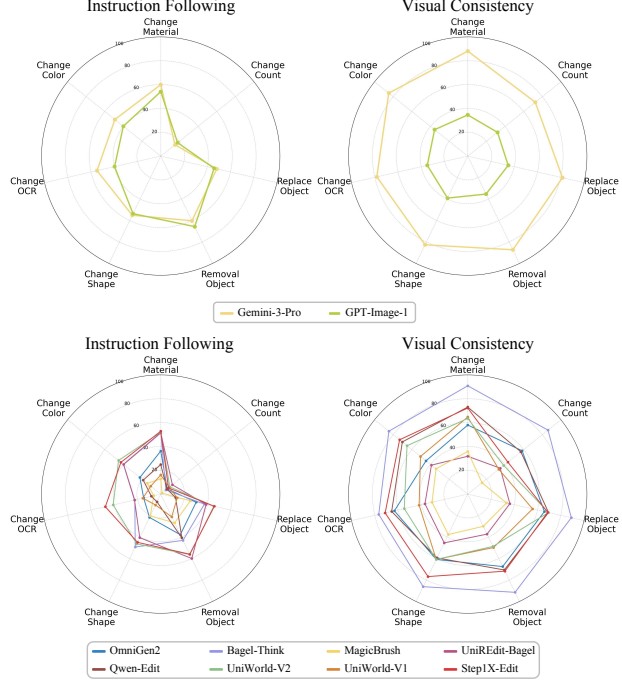

*Figure 7.* Performance on DLEBench across different instruction types and two different criteria. Top: Closed-source models. Bottom: Open-source models.

making it the most challenging type in our benchmark.

**Analysis by Evaluation Criteria.** Figure 7 presents a radar chart depicting model performance across various criteria, with specific values detailed in Table 4. Regarding IF, the results indicate that all models perform suboptimally. Even the top-performing Gemini-3-Pro averages only 48.97, un-

*Table 2.* IAA on DLEBench. We report Krippendorff's Alpha ($\alpha$) for both IF and VC across different instruction types.

| Instruction Types | Numbers | IF ($\alpha$) | VC ($\alpha$) |
|---|---|---|---|
| Change Material | 84 | 84.43 | 85.65 |
| Change Color | 699 | 85.41 | 92.12 |
| Change OCR | 455 | 83.80 | 89.62 |
| Change Shape | 23 | 81.77 | 95.41 |
| Removal Object | 577 | 93.32 | 94.60 |
| Replace Object | 47 | 87.64 | 94.49 |
| Change Count | 24 | 79.57 | 92.12 |
| Overall | 1889 | 90.23 | 92.24 |

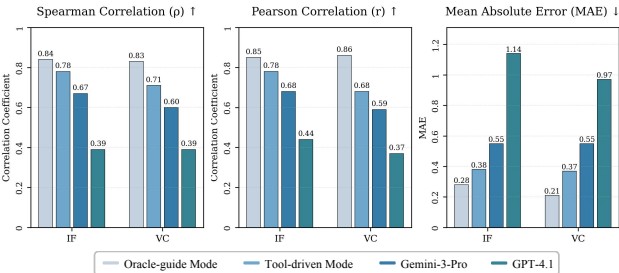

*Figure 8.* Correlation coefficients and MAE between human judgments and different evaluation methods across IF and VC.

derscoring a significant deficiency in the small-scale editing abilities of existing IIEMs. To visually illustrate these limitations, we provide extensive qualitative comparisons in Appendix C. Conversely, most models perform well on VC, which assesses global consistency. However, a striking disparity exists between GPT-Image-1 and Bagel-Think (35.17 vs. 86.43). We attribute this to GPT-Image-1's tendency to erroneously modify non-target regions when it fails to localize the target objects, thereby severely compromising global consistency. In contrast, although Bagel-Think yields a lower IF score than GPT-Image-1 (35.55 vs. 45.32), it avoids indiscriminate modifications during localization failures. These findings suggest that a conservative strategy—abstaining from editing under uncertainty—is preferable to aggressive modification for preserving global consistency.

**Impact of Object Scale on Performance.** To investigate the sensitivity of IIEMs to target object scale, Figure 6 visualizes the relationship between the target object's bounding box area and the IF score. The scatter plots use logarithmic axes for relative width and height, clustering small-scale objects in the lower-left quadrant and large-scale ones in the upper-right. Two distinct behavioral patterns emerge from these results. For most models, performance shows a scale-dependent pattern: target objects with small spatial occupancy are associated with low scores (red points), whereas larger objects are associated with high scores (blue points). In contrast, models like UniWorld-V1, Qwen-Edit,

and MagicBrush show consistently poor performance across all scales, evidenced by the dominance of low-scoring samples (red points) throughout the spectrum. Consequently, compared to other IIEMs, these models lack the ability to edit small-scale objects.

We further quantify these visual observations through the correlation analysis presented in the top-left inset. Specifically, we apply a sliding window (size=10) to calculate average scores across samples sorted by bounding box area. While most competitive models show a positive correlation—confirming they are effectively constrained by object scale—the three aforementioned models display notably weak correlations. The lack of correlation suggests that the editing tasks exceed the models' capabilities, even at the largest object scales in our benchmark.

Finally, the score distribution (bottom-right inset) provides granular insights into the nature of model failures. A predominance of Score 1 (Red: Localization Failure) in models such as OminiGen2, UniWorld-V1, Qwen-Edit, MagicBrush, and Bagel-Think underscores a critical deficiency in identifying target objects. Conversely, Step1X-Edit, UniWorld-V2, and Gemini-3-Pro exhibit a high frequency of Score 3 (Blue: Over-Modification). These results suggest that while these models accurately locate targets, they tend towards excessive alteration, compromising attribute preservation. Distinctly, the prevalence of Score 2 (Orange: Wrong Action) in GPT-Image-1 indicates that its primary bottleneck lies in instruction adherence rather than spatial localization. Notably, UniREdit-Bagel displays systemic shortcomings, underperforming across localization, execution, and preservation metrics.

### 5.3. Validity of Dual-mode Evaluation Framework

To validate our framework, we assess its alignment with human judgments. Specifically, four annotators evaluate a randomly sampled subset of the outputs generated by the 10 IIEMs across the entire DLEBench (Annotation Document in Appendix A). Before formal annotation, we conducted a calibration phase on 100 randomly sampled outputs to resolve discrepancies arising from ambiguous cases and unify evaluation criteria, thereby maximizing Inter-Annotator Agreement (IAA). With these annotations serving as the ground truth, we quantify alignment using Spearman ($\rho$) and Pearson ($r$) correlation coefficients, alongside Mean Absolute Error (MAE). In terms of implementation, we use Gemini-3-Pro as the backbone LMM for both our evaluation framework and the LMM-as-a-Judge, while employing GPT-4.1 as the evaluator within the LMM-as-a-Judge paradigm for reference. All prompts are detailed in Appendix G. To ensure fairness, we use an identical base prompt, appending tool instructions only for methods that require them.

In Table 2, we present the IAA calculated using Krippen-

dorff's Alpha ($\alpha$) across various instruction types. Guided by our rigorous annotation guidelines, we observe high consistency scores across all categories. These results attest to the reliability of the human annotations, providing a solid foundation for validating our evaluation framework.

Building on this reliable human baseline, we quantify the alignment between different evaluation methods and human judgments in Figure 8. The results show that the Oracle-guided Mode yields the strongest agreement, achieving the highest $\rho$ and $r$ along with the lowest MAE, followed by the Tool-driven mode. In contrast, the LMM-as-a-Judge baselines, whether employing Gemini-3-Pro or GPT-4.1, exhibit lower correlation coefficients and significantly higher MAE. This performance gap indicates that even SOTA LMMs struggle to reliably evaluate our benchmarks (More qualitative comparisons in Appendix F).

## 6. Conclusion, Limitation and Future Work

In this paper, we introduce DLEBench, a benchmark with 1,889 samples for evaluating the small-scale object editing ability of IIEMs, where target objects occupy just 1%–10% of the area. Evaluating 10 IIEMs, we find that current models struggle to maintain editing fidelity and consistency under such constraints. To enable reliable assessment, we propose a dual-mode evaluation framework with refined rubrics that better align with human judgments than LMM-as-a-Judge.

A primary limitation of our benchmark is the unbalanced distribution of instruction types, stemming from the conversion of samples from visual reasoning datasets. As a result, certain editing operations are underrepresented, and the benchmark currently lacks more complex manipulations such as moving, scaling, and rotating objects, as well as multi-instruction coupling scenarios that are common in real-world image editing workflows. In future work, we will develop an automated data expansion pipeline to improve both the diversity and balance of the instruction distribution, while incorporating a broader range of complex editing operations and compositional instruction patterns.

## Acknowledgements

This project was supported by the National Natural Science Foundation of China (NSFC) under Grant No. 62576102.

## Impact Statement

This paper presents work whose goal is to advance the field of Machine Learning. There are many potential societal consequences of our work, none of which we feel must be specifically highlighted here.

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

# A. Annotation Document

You will be presented with a Source Image, an Editing Instruction, an Edited Image, and a Reference Image. Your task is to evaluate the quality of the edit based on two distinct dimensions: Instruction Following and Visual Consistency. The annotation interface is shown in Figure 11.

## A.1. Definitions of Criteria

**Instruction Following:** This dimension assesses whether the model accurately localizes the target object and correctly executes the edit while strictly preserving intrinsic attributes irrelevant to the instruction, serving as a primary metric for evaluating small-scale editing ability. Given that the target regions or objects occupy a small pixel ratio, may be partially occluded, or appear blurred, the evaluation focuses on the model's ability to precisely locate the target region, while ensuring that excessive modifications are not made to the target object. It is necessary to consider the following three sub-dimensions:

- Target Object Location: Did the model perform the modification at the correct target location?

- Action Alignment: Did the model perform the correct modification action (e.g., replace, remove, alter) on that object as requested?

- Over Modification: Did the model preserve the original identity and details of the target object that were not specified to change?

**Visual Consistency:** This dimension measures how well the environments and visual elements unrelated to the instruction are preserved between the input and output images, which considers global preservation. This is particularly important in visual editing tasks, as it distinguishes between models that perform grounded edits based on the original image (e.g., native generation models) and those that regenerate scenes from scratch (e.g., cascade-based models). It is necessary to consider the following three sub-dimensions:

- Global Scene Stability: Did the model preserve the fundamental visual context of the scene, including artistic style and scene style?

- Local Anomaly Detection: Did the model strictly preserve the non-target objects, ensuring no changes, deletions, distortions, or additions of extra objects?

## A.2. Score Rubrics

Tables 5 and 3 present the score rubrics for Instruction Following and Visual Consistency, respectively. The Instruction Following comprises four hierarchical levels, ranging from best to worst: Score 4 (Flawless Execution), Score 3 (Over Modification), Score 2 (Wrong Action), and Score 1 (Localization Failure). Meanwhile, Visual Consistency comprises four hierarchical levels: Score 4 (Perfect Consistency), Score 3 (Single Anomaly), Score 2 (Multiple Anomalies), and Score 1 (Scene Collapse). We employ discrete adjective ratings rather than numerical scores to facilitate better understanding by the LMMs.

## A.3. Evaluation Process

We present the evaluation pipeline for both criteria, along with two annotation examples as shown in Figure 9.

**For Instruction Following**, the following evaluation process can be applied:

Step 1: Check for Precise Localization

- Compare the Source Image and the Edited Image. Do not rely on mere visual differences or pixel-level changes. Instead, focus on whether the intended edit has effectively occurred. Even if the image exhibits changes (such as artifacts, color shifts, or slight distortions), if the target object remains identifiable or has not undergone the requested modification, you must label it as Localization Failure.

- If severe blurriness makes it impossible to distinguish whether a modification occurred, or if the modification explicitly affects regions outside the target area, stop and label as Localization Failure.

*Table 3.* Scoring rubrics for the Visual Consistency.

| Label | Description |
|---|---|
| Perfect Consistency | The highest standard. The background environment and all non-target objects remain visually identical to the Original Image. The edit integrates seamlessly without disturbing the surrounding pixels. The scene looks like the exact same photo, just with the specific target modified. |
| Single Anomaly | The general background environment remains consistent, but exactly ONE specific non-target object or detail has been altered, removed, or distorted. For example, everything is perfect except for one cup on the table that changed color, or the addition of one person who did not exist in the original image. Changes that only affect the overall image texture or global visual effects (e.g., lighting filters or grain) are not considered anomalies. |
| Multiple Anomalies | The general background environment remains consistent, but TWO OR MORE distinct non-target objects or details have been altered, removed, or distorted. For example, a painting on the wall changed content AND a chair in the corner disappeared. There are multiple scattered errors in the scene. Similarly, changes that only affect the overall image texture or global visual effects are not counted as anomalies. |
| Scene Collapse | The high-level semantic category of the environment or the artistic style has fundamentally changed. The Edited Image depicts a completely different type of location (e.g., a city turning into a forest) or implies a complete shift in medium (e.g., from photorealistic to an oil painting). Crucial: You must IGNORE changes to specific background objects or the layout of the scene; this label applies strictly when the general setting category or artistic style is destroyed. |

– If the model performs any modification on an incorrect sub-component within the target object, and the specific part or attribute specified in the instruction is not correctly modified, label it as Localization Failure.

– If a change occurred, proceed to Step 2.

Step 2: Check the Action Alignment

– Compare the Edited Image with the Source Image. Did the model perform the correct action requested in the text?

– If a change occurred, check if it matches the Editing Instruction. Determine if the specific action (e.g., color change, object removal) was performed correctly. You may consult the Reference Image for visual context, but the text instruction is the primary rule. If the model performed a modification that contradicts the instruction (e.g., turning an object green instead of red), label it as Wrong Action.

– For object count reduction instructions, if the target objects are not reduced to the exact specified quantity, this is classified as Wrong Action; similarly, for object addition instructions, if the target objects are not increased to the exact specified quantity, this is also classified as Wrong Action.

– If the action was performed correctly, proceed to Step 3.

Step 3: Check for Over Modification

– Compare the Edited Image with the Source Image. Did the object retain its original shape, texture, and structural details consistently, except for the parts explicitly targeted by the instruction?

**Editing Instruction 1:** Changed color of the red bus to yellow.

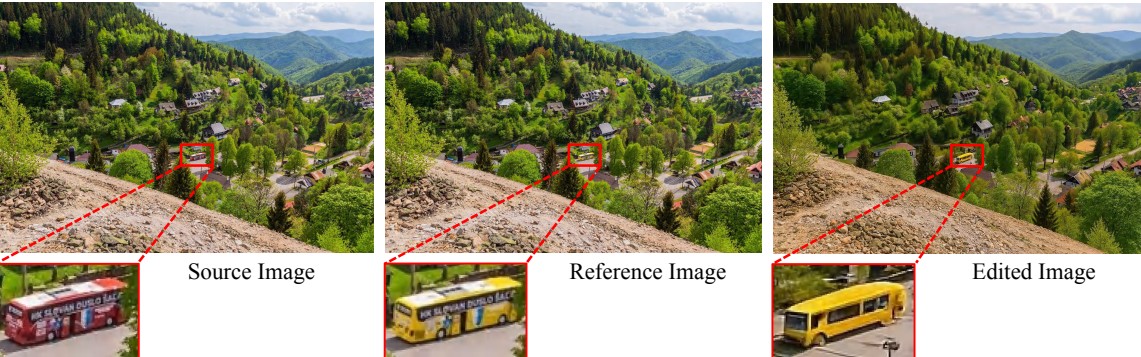

**Criterion:** Instruction Following
**Evaluation Result:** Over Modification
**Reasoning:** The model correctly located the red bus and changed its color to yellow. However, the details of the bus have been altered, including changes to the text on the windows and the icons on the body.

**Editing Instruction 2:** Changed the attribute of the woman's scarf from red to green.

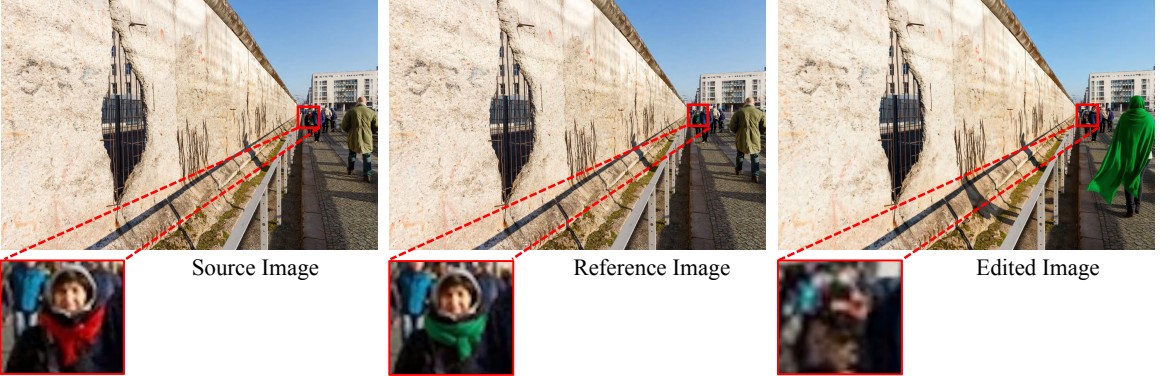

**Criterion:** Visual Consistency
**Evaluation Result:** Multiple Anomalies
**Reasoning:** Without considering the modifications to the red target area, the image style of the Edited Image remains consistent with that of the Source Image, and there is no excessive blurring. However, the Edited Image changed the man in beige clothing on the right into a woman wearing a green cloak, which introduced one error. Also, the person next to the woman has severe distortion, which is also an anomaly.

*Figure 9.* Annotation examples for the two evaluation dimensions.

– If the object is structurally distorted, has unrequested style changes (e.g., a T-shirt becoming a Hoodie), label as Over Modification. Note: For object removal instruction, the deletion of non-target objects does not count as Over Modification. For object replacement instructions, if the target is successfully replaced with the requested object type and the replaced object itself is clearly recognizable as that object, alterations to surrounding objects are not classified as Over Modification. However, if the replaced object is not clearly recognizable, it should be classified as Over Modification.

– If the modification action is recognizable, but the other parts of the object have become blurry or lost fine details, preventing you from confirming its original shape, texture, and structure, label as Over Modification.

– If the object remains sharp and faithful to the original, label as Flawless Execution.

**For Visual Consistency**, the following evaluation process can be applied:

Step 1: Check for Global Scene Stability

– Compare the background environment of the Edited Image with the Source Image, strictly ignoring the target objects.

*Table 4.* Performance on DLEBench across different instruction types and two different criteria.

| Instruction Types | Closed-Source Models | | Open-Source Models | | | | | | | |
|---|---|---|---|---|---|---|---|---|---|---|
| | Gemini-3-Pro | GPT-Image-1 | OmniGen2 | Bagel-Think | UniREdit-Bagel | MagicBrush | Qwen-Edit | UniWorld-V1 | UniWorld-V2 | Step1X-Edit |
| **Instruction Following (IF)** | | | | | | | | | | |
| Change Material | **59.92** | 54.07 | 36.11 | 51.19 | 51.59 | 12.70 | 24.80 | 16.06 | 51.98 | 52.78 |
| Change Color | **49.21** | 40.17 | 22.37 | 39.82 | 39.97 | 14.62 | 18.84 | 10.70 | 45.27 | 42.74 |
| Change OCR | **54.84** | 39.73 | 14.73 | 22.34 | 22.56 | 6.04 | 8.09 | 15.45 | 40.95 | 47.69 |
| Change Shape | **55.07** | 53.62 | 21.74 | 49.28 | 40.58 | 20.29 | 7.25 | 10.14 | 46.38 | 44.93 |
| Removal Object | 60.26 | **65.68** | 37.82 | 43.09 | 60.01 | 26.80 | 40.76 | 21.20 | 56.32 | 56.00 |
| Replace Object | **48.23** | 45.93 | 30.50 | 37.59 | 39.01 | 25.53 | 12.77 | 14.19 | 45.65 | 46.10 |
| Change Count | 15.28 | **18.06** | 6.94 | 5.56 | 12.50 | 1.39 | 6.67 | 9.11 | 9.72 | 8.33 |
| Average | **48.97** | 45.32 | 24.32 | 35.55 | 38.03 | 15.34 | 17.03 | 13.84 | 42.32 | 42.65 |
| **Visual Consistency (VC)** | | | | | | | | | | |
| Change Material | 88.10 | 34.54 | 57.94 | **90.87** | 31.73 | 35.77 | 73.02 | 64.68 | 63.49 | 72.22 |
| Change Color | **84.76** | 35.57 | 44.79 | 84.65 | 38.97 | 34.01 | 70.20 | 50.67 | 65.14 | 73.11 |
| Change OCR | **78.23** | 34.82 | 63.13 | 76.58 | 36.82 | 30.73 | 65.49 | 41.85 | 54.63 | 71.09 |
| Change Shape | 82.61 | 39.13 | 60.87 | **86.21** | 45.45 | 37.68 | 59.42 | 60.87 | 60.87 | 76.81 |
| Removal Object | 87.25 | 35.36 | 67.51 | **91.49** | 37.28 | 30.01 | 70.68 | 49.79 | 48.73 | 71.80 |
| Replace Object | 81.49 | 34.81 | 65.96 | **89.13** | 36.23 | 33.33 | 69.50 | 55.80 | 69.50 | 68.79 |
| Change Count | 72.46 | 31.94 | 58.33 | **86.11** | 34.85 | 15.28 | 56.94 | 33.33 | 38.56 | 43.06 |
| Average | 82.13 | 35.17 | 59.79 | **86.43** | 37.33 | 30.97 | 66.46 | 51.00 | 57.27 | 68.13 |

- Did the model preserve the general background environment and artistic style?

- If the scene looks like a completely different place, a different time of day, or has a shifted style, stop and label as Scene Collapse.

- If the global environment remains stable, proceed to Step 2.

Step 2: Scan for Local Anomalies

- Meticulously scan the non-target areas for specific objects or details that appear in the Source Image but are missing, distorted, or changed in the Edited Image, as well as any new objects or elements appearing in the Edited Image that were not present in the Original Image.

- Count the number of distinct errors found (e.g., a missing lamp, a changed rug pattern).

- If you find **two or more** distinct errors, label as Multiple Anomalies.

- If you find exactly **one** distinct error, label as Single Anomaly.

- If the background is pristine with **zero** errors, label as Perfect Consistency.

**Note on Blurring**: If the target is too blurry to verify whether the localization was successful, label it as Localization Failure. If the edited object clearly follows the editing instruction, but other parts of the object appear blurred or degraded, preventing you from judging whether original features were preserved, label it as Over Modification.

**Note on Multiple Target Objects**: The target object may appear multiple times in an image. In such cases, each target instance must be evaluated independently following the above criteria, and the final score for the image should be the worst result among all target objects.

## B. More Quantitative Results

Table 4 presents the performance of ten IIEMs across two dimensions and seven instruction types. We utilize oracle-guided mode to ensure assessment accuracy and normalize all scores to a 100-point scale to facilitate direct cross-model comparison.

## C. More Qualitative Comparison Results

We provide qualitative comparisons on DLEBench between Figure 13 and 19

*Table 5.* Score rubrics for the Instruction Following.

| Label | Description |
|---|---|
| Flawless Execution | The model demonstrates flawless instruction following. It correctly identifies the target region or object, executes the specific modification accurately, and strictly preserves the object's original identity (including shape, texture, and structure) without introducing any unintended changes. |
| Over Modification | This label applies when the model correctly locates the target and executes the requested edit, but fails to preserve the object's original details that should have remained unchanged. Except for the specific changes explicitly requested by the instruction, the target object must remain visually identical to the Original Image; therefore, any unrequested alterations to the object's structure, or fine details—such as changing a T-shirt into a hoodie when only a color shift was requested—are classified as Over Modification. Notably, for object removal instructions, the accidental deletion of non-target objects is NOT penalized under this category. For object replacement instructions, as long as the target is successfully replaced with the requested type of object and the replaced object itself is clearly recognizable as that object, alterations to surrounding objects are NOT classified as Over Modification. However, if the replaced object is not clearly recognizable, it should be classified as Over Modification. Additionally, if the new object is inconsistent with the original image's style (e.g., lighting, rendering style, or realism level), it should also be classified as Over Modification. |
| Wrong Action | The model successfully locates the target object, but executes a modification that mismatches the requested action category. You must strictly verify against these specific categories: Change Color, Change Material, Change Text, Change Shape, Object Count (Reduction, Addition), and Object Manipulation (Remove, Replace). If the model performs an action from a different category than requested (e.g., executing a "Remove Object" operation when a "Change Color" was requested, or "Replace Object" when only a "Change Material" was asked), it must be labeled as Wrong Action. Notably, for object count reduction instructions, if the target objects are not reduced to the exact specified quantity, this is classified as Wrong Action; similarly, for object addition instructions, if the target objects are not increased to the exact specified quantity, this is also classified as Wrong Action. For the object removal instructions, filling in the area where the object was located after removal shall not be considered a Wrong Action, unless it is clearly evident that a replacement operation has been performed. |
| Localization Failure | Localization Failure occurs when the model fails to execute the specified modification on the intended target, leaving the object effectively unchanged. This score encompasses scenarios where the target exhibits minor, unintentional artifacts—such as subtle color shifts or slight geometric distortions—that do not constitute the requested edit, as well as cases where severe blurriness or distortion renders visual verification of the modification impossible. Furthermore, this score also includes mis-localization within the target object, where the model modifies an incorrect sub-component of the target object (e.g., when the target object is a wheel and the instruction is to change the hub of the wheel to red, but the model instead changes the tire to red), because the specific attribute defined by the instruction remains unmodified. |

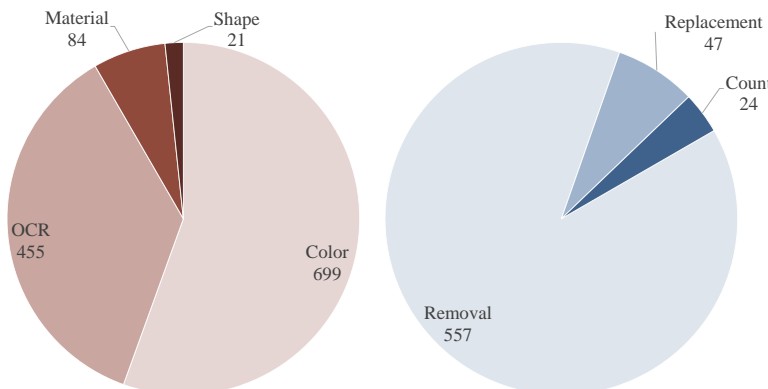

(a)Attribute-level modification categories    (b)Object-level modification categories

*Figure 10.* Instruction Statistics of DLEBench. (a) Distribution of attribute-level modifications, dominated by color and OCR. (b) Distribution of object-level modifications, where removal is the most frequent operation.

## D. Distribution of Editing Instruction Types

Through our data transformation pipeline, DLEBench comprises 1,889 high-quality image editing samples spanning seven distinct editing instruction types, categorized into attribute-level and object-level modifications. Figure 10(a) illustrates the distribution of attribute-level modifications. Inheriting the characteristics of the source visual reasoning benchmarks, Color and OCR emerge as the predominant categories, as these visual properties are most frequently queried in visual reasoning tasks. Beyond these dominant types, the dataset encompasses other attribute-based edits, including Material, Shape. Figure 10(b) details object-level modifications. Similarly, influenced by the prevalence of object existence queries in the source data, Removal accounts for the majority of instances, whereas Replacement and Count appear less frequently.

## E. Calculation of Target Area Ratios

To compare target scale distributions against mainstream benchmarks (ImageEdit (Ye et al., 2025), UniREditBench (Han et al., 2025), RISE (Zhao et al., 2025), and KRIS-Bench (Wu et al., 2025c)), we address the lack of ground truth bounding boxes of target objects in these datasets using a two-stage pipeline. First, we utilize GPT-4.1 to extract the specific target object name from each editing instruction, following the prompts detailed in Table 6. Second, leveraging the observation that targets in these benchmarks are typically prominent, we employ GroundingDINO to localize objects and generate bounding boxes from the extracted names, enabling the calculation of target area ratios.

*Table 6.* The prompt used by GPT-4.1 to extract target objects.

> You are an expert in semantic analysis for image editing tasks. Your goal is to extract the **primary target object** that needs to be modified, removed, or replaced based on the user's editing instruction.
> **Rules:** 1. Identify the specific object being acted upon. 2. If the instruction implies the whole image (e.g., "make it look cinematic"), output "image". 3. If the instruction is to replace Object A with Object B, the target is **Object A** (the original object). 4. Output **only** the object name (noun or noun phrase), without unnecessary articles (a, an, the) or excessive adjectives unless necessary to distinguish the object. 5. Strictly follow the output format: '[Result]: <Object Name>'
> **Examples:**
> Input: Change the building's exterior color to a light beige. Output: [Result]: building
> Input: Remove the person standing on the left. Output: [Result]: person
> Input: Replace the cat with a dog. Output: [Result]: cat
> Input: Make the red car look like a vintage car. Output: [Result]: red car
> Input: Add a smile to the woman's face. Output: [Result]: woman's face
> **Current Input:** INPUT_INSTRUCTION
> **Output:**

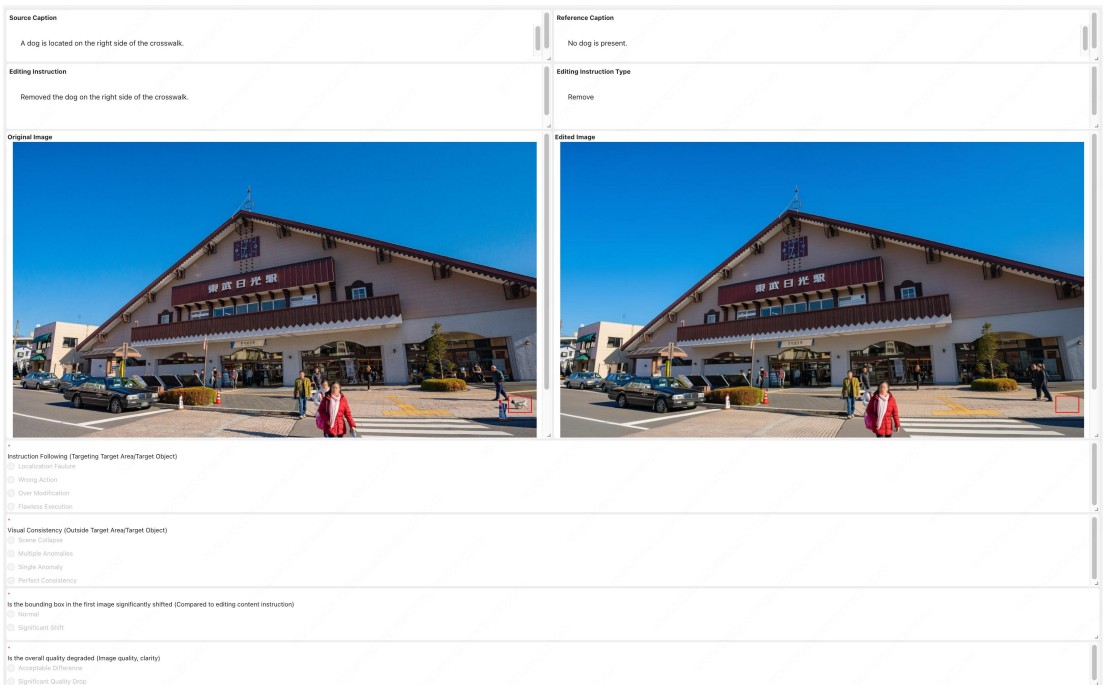

*Figure 11.* The user interface for annotating IF and VC for human judgements.

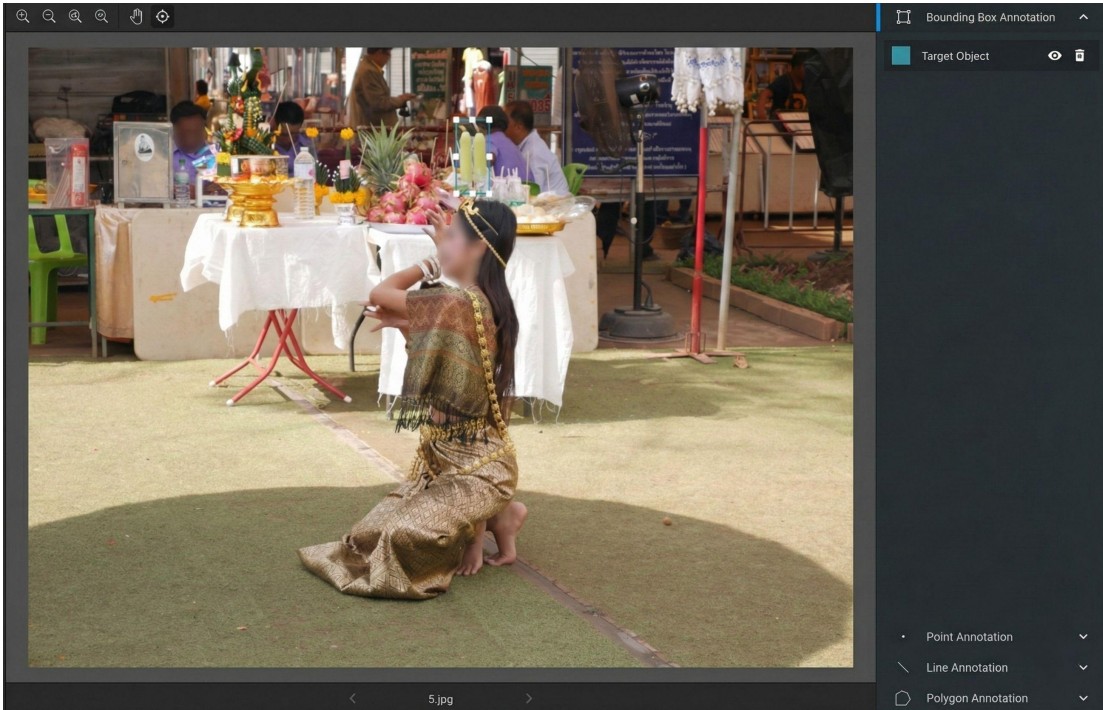

*Figure 12.* The user interface for labeling the target objects' bbox.

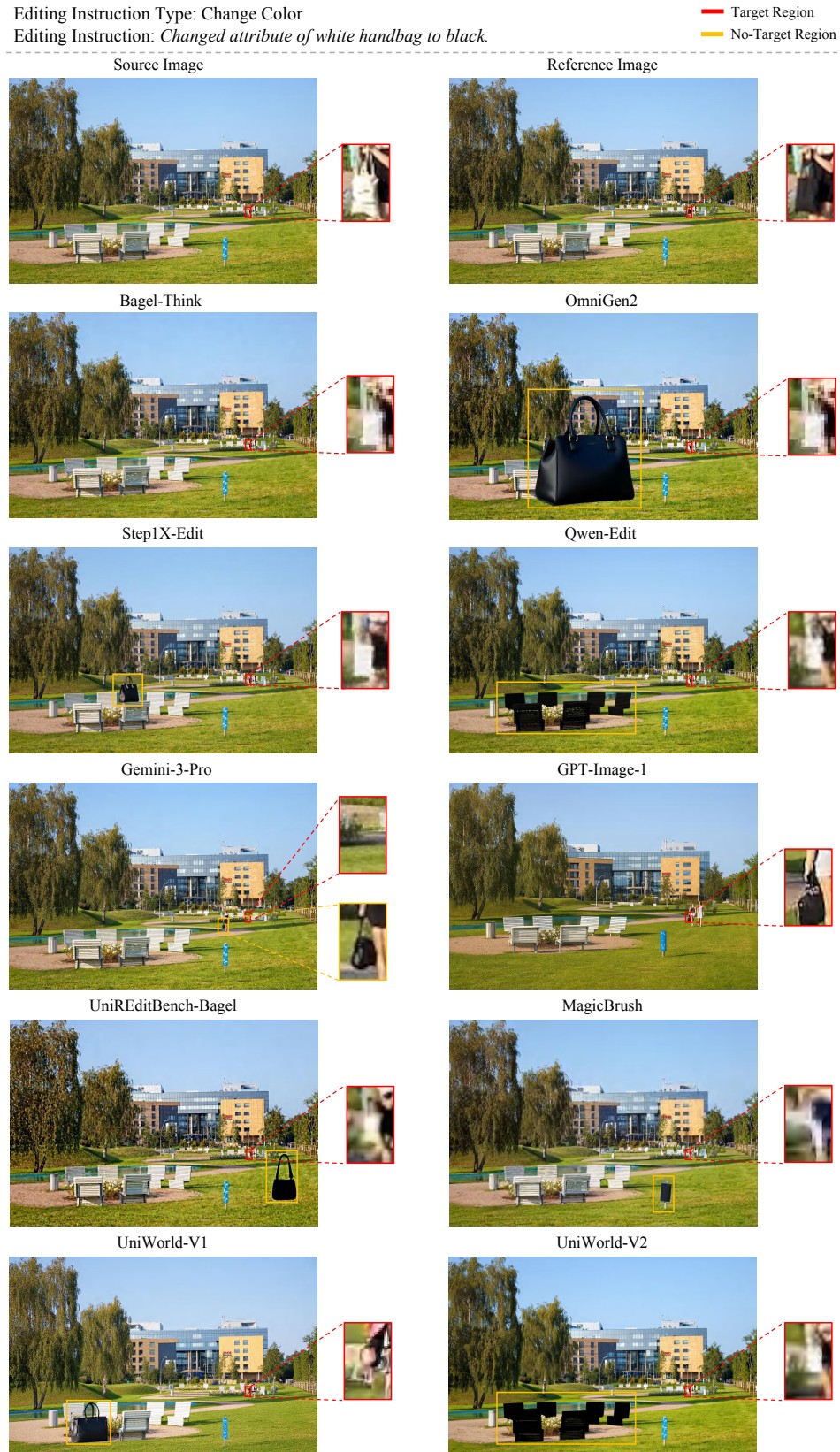

*Figure 13.* Visualization results for Change Color.

Editing Instruction Type: Change OCR
Editing Instruction: *Change the text content on the white bus from 'TR 7906 T' to 'TR 9067 T'.*

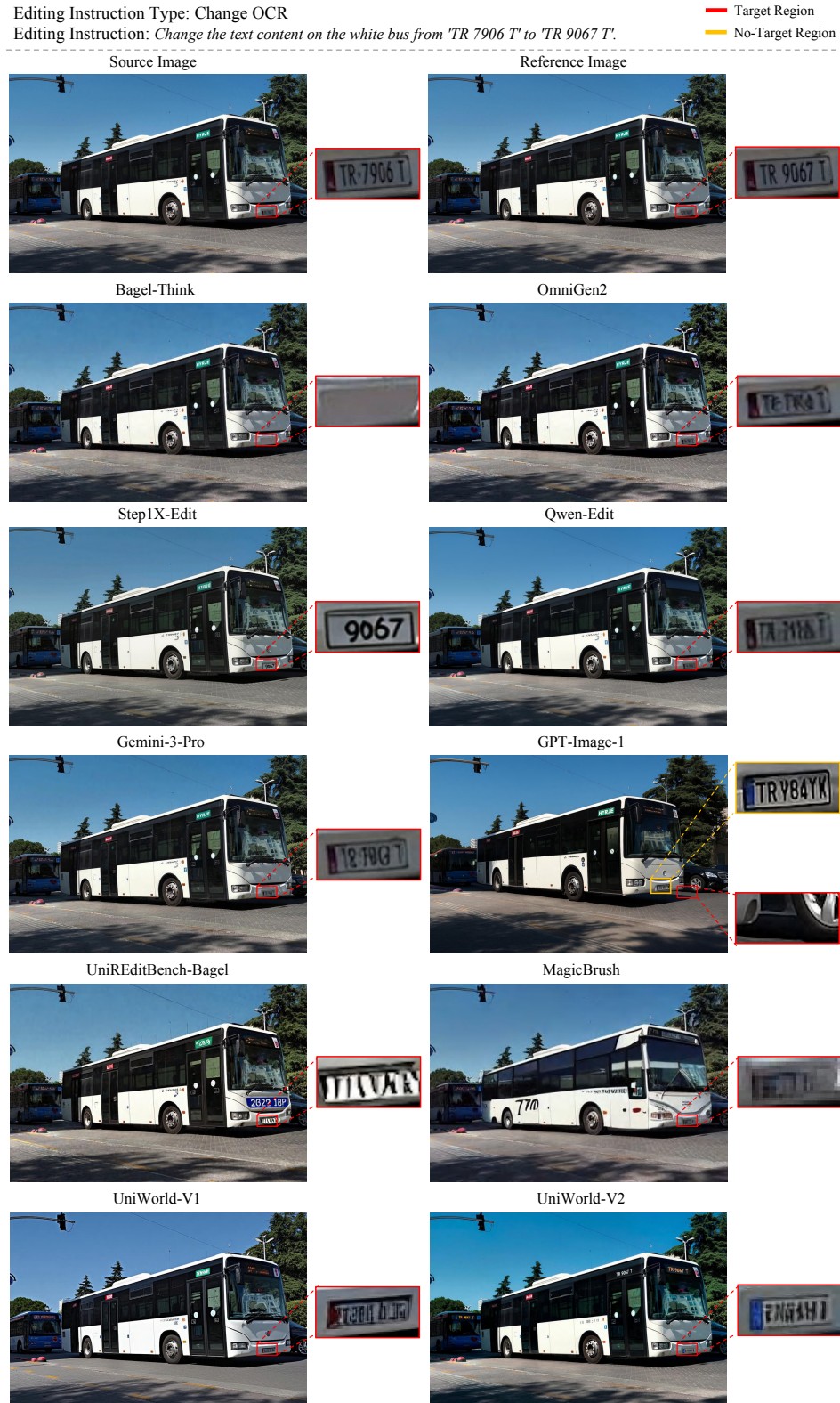

*Figure 14.* Visualization results for Change OCR.

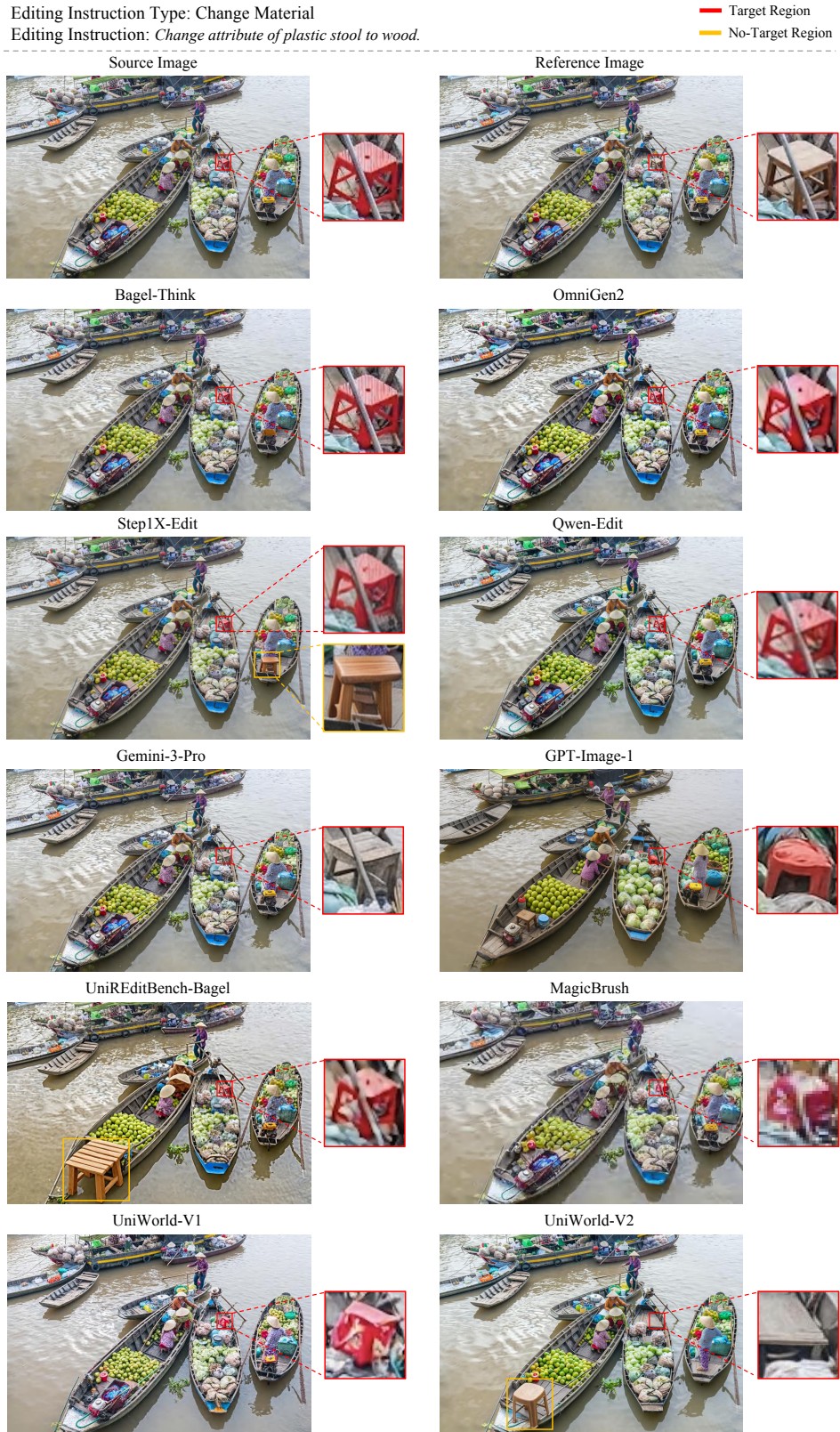

*Figure 15.* Visualization results for Change Material.

Editing Instruction Type: Replace Object
Editing Instruction: *Change spider to dog on the blue sail.*

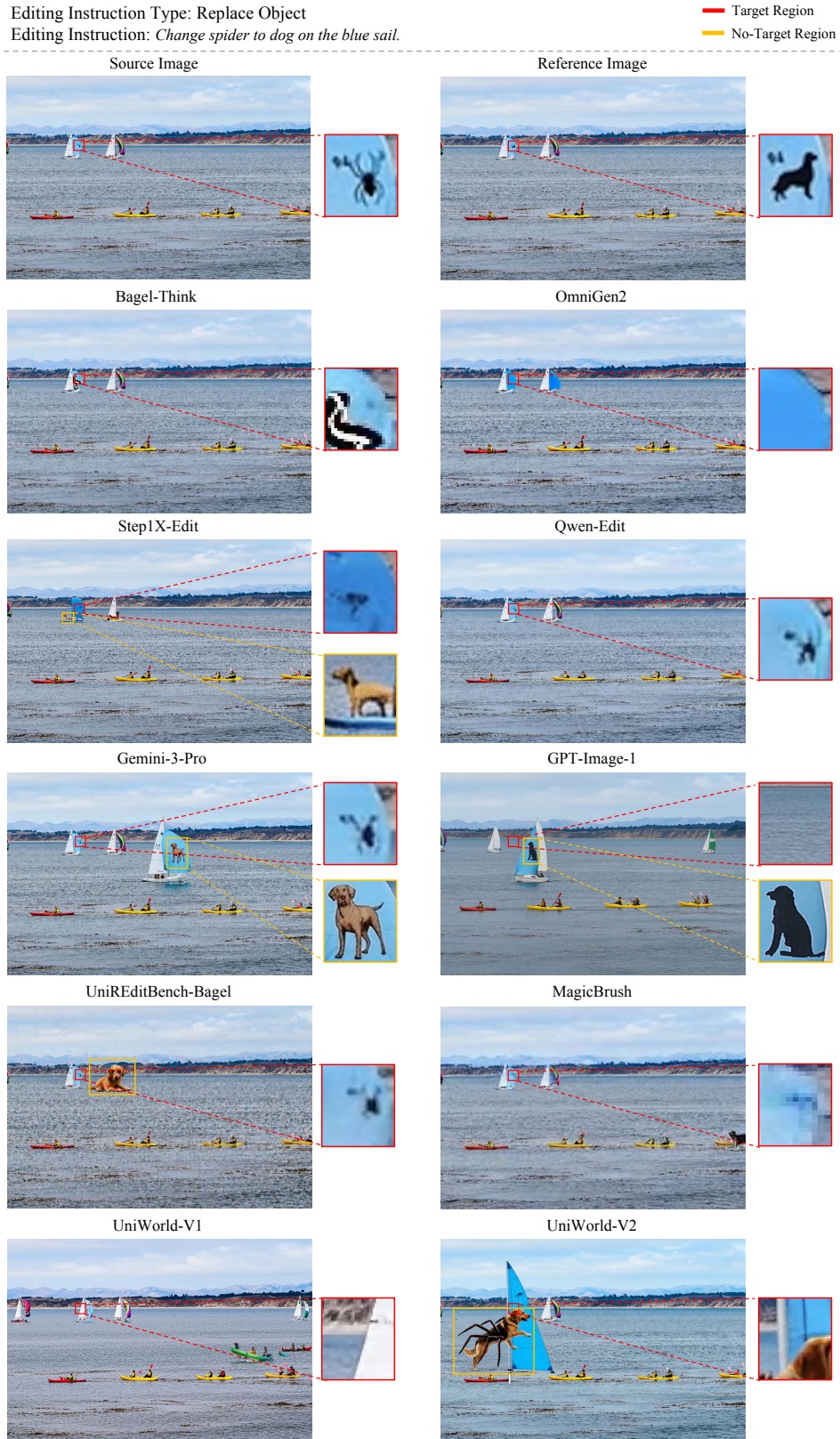

*Figure 16.* Visualization results for Replace Object.

Editing Instruction Type: Change Count
Editing Instruction: *Change the count of people in the oil painting from 2 to 4.*

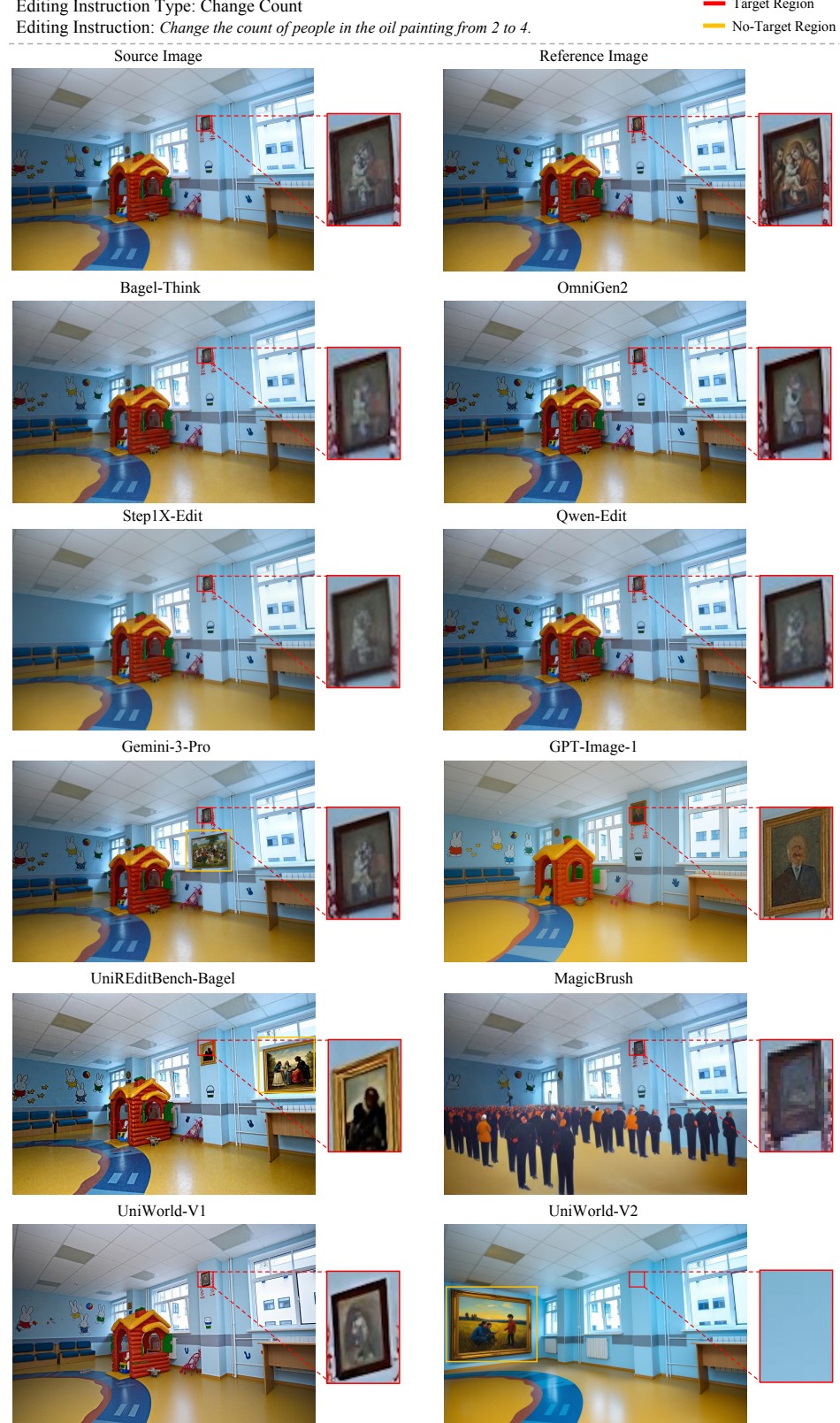

*Figure 17.* Visualization results for Change Count.

Editing Instruction Type: Removal Object
Editing Instruction: *Remove the telephone located on the right side of the hand lamp.*

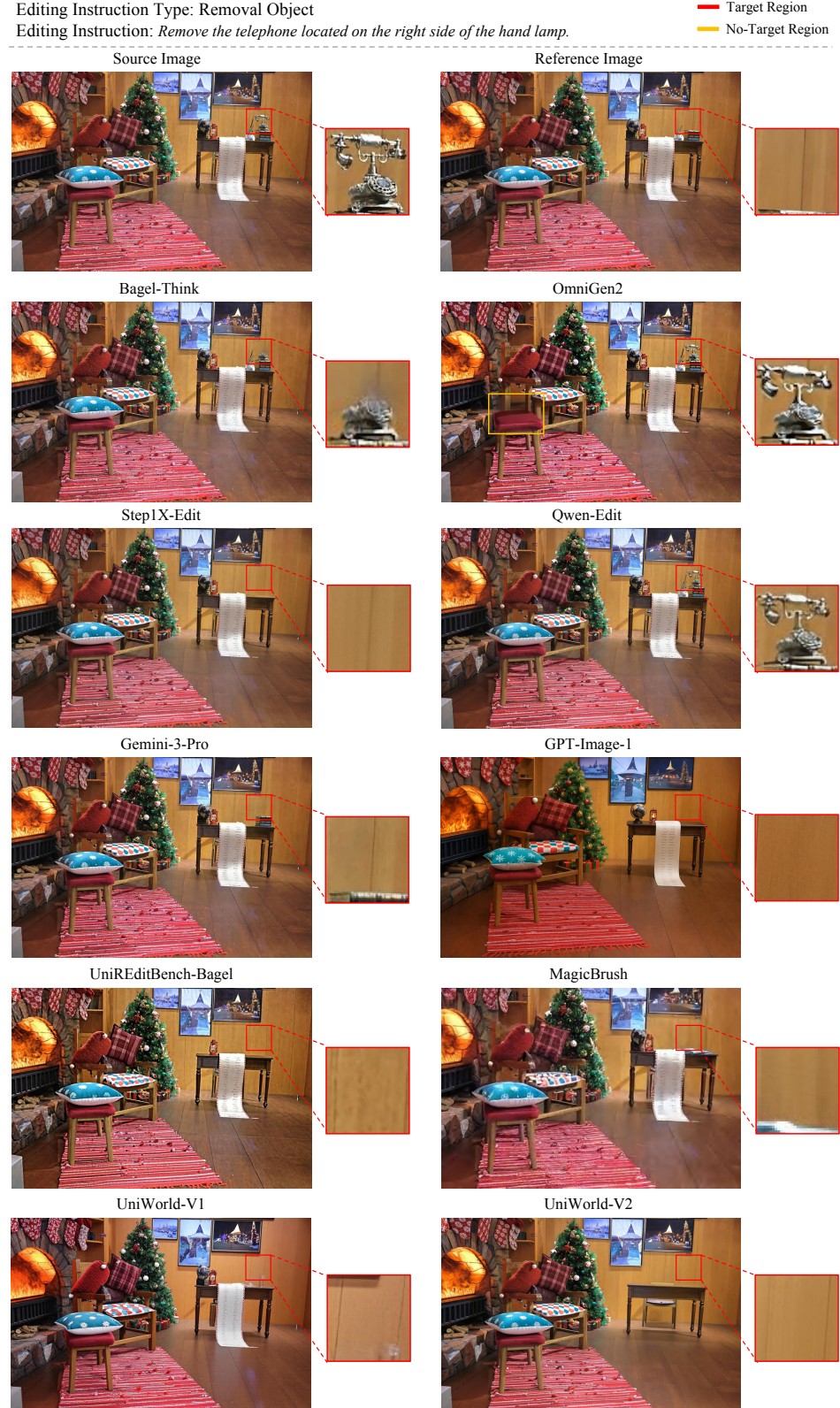

*Figure 18.* Visualization results for Removal Object.

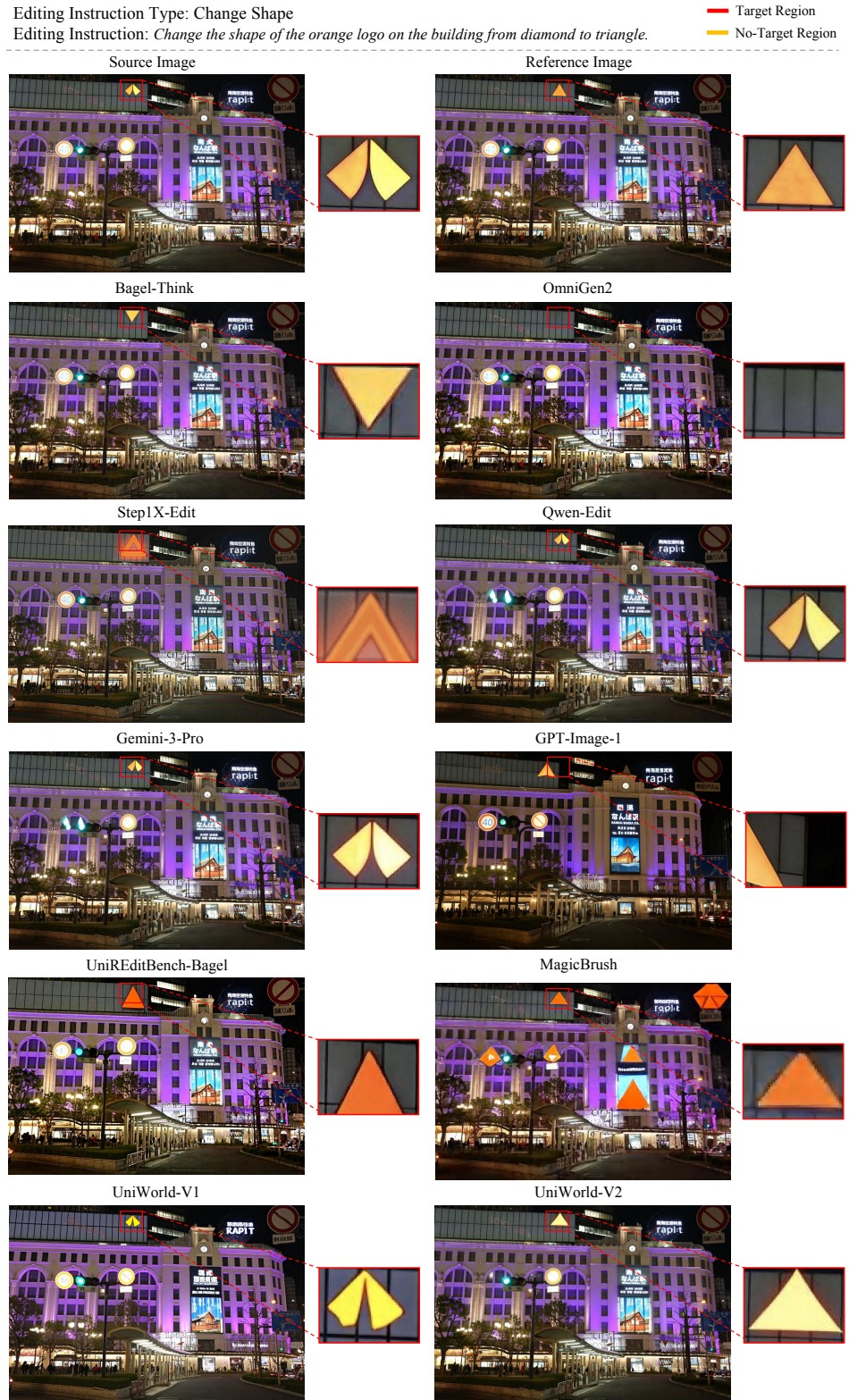

*Figure 19.* Visualization results for Change Shape.

## F. The Qualitative Comparison Example of Evaluation Method

Tables 7 to 10 compare our Oracle-guided and Tool-driven Modes against Gemini-3-Pro and GPT-4.1 baselines on IF criterion. The results demonstrate that the Oracle-guided Mode leverages human-annotated bbox to crop the target object in advance, enabling the evaluator to focus its attention specifically on the relevant region. Additionally, the Tool-driven Mode utilizes external tools to localize differences, generating visual comparisons that highlight discrepancies in both the target object and non-target areas between the source and edited images. Notably, both of our proposed modes successfully identified that the model, while modifying the color, inadvertently altered the shape of the dustpan. This resulted in a verdict of "Over Modification," which aligns with human evaluation. In contrast, both Gemini-3-Pro and GPT-4.1 failed to perceive the shape deformation alongside the color change due to limited visual perception capabilities, leading to an erroneous assessment of "Flawless Execution".

Similarly, Tables 11 to 14 present the comparative results of different methods on VC criterion. Since both the Oracle-guided and Tool-driven Modes can use external tools for VC evaluation, they effectively detect discrepancies in non-target regions, yielding a verdict of "Multiple Anomalies" that aligns with human assessment. In contrast, Gemini-3-Pro identified only one discrepancy, concluding with "Single Anomaly", while GPT-4.1 failed to detect any anomalies, resulting in an erroneous assessment of "Perfect Consistency".

*Table 7.* An example of the feedback generated by Oracle-guide Mode in Instruction Following.

---

**Example 1.** Using Oracle-guide Mode to Evaluate an Edited Image Generated by UniWorld-V2

**Editing Instruction:** Change attribute of blue dustpan to red.

| **Source Image** | **Edited Image** | **Reference Image** |
|:---:|:---:|:---:|
| 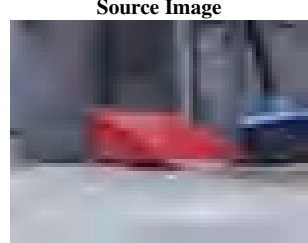 | 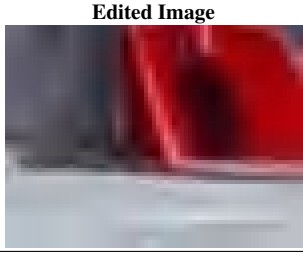 | 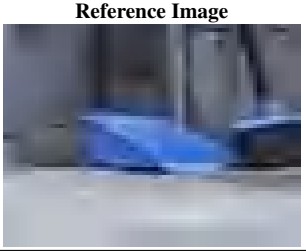 |

Gemini-3-Pro

⟨**Start Thinking**⟩

1. **Verify Modification Occurrence**: I compare the Original Image with the Edited Image. The Original Image displays a blue dustpan with a tall black handle standing next to a broom. The Edited Image shows a large red shovel-like object with a red handle. There is a distinct visual difference, so a modification has definitely occurred. 2. **Verify Action Alignment**: The Editing Instruction is "Changed attribute of blue dustpan to red". This is a request to change the color of the dustpan. The Edited Image shows an object that is now red. Thus, the model attempted the correct action category (changing color) and achieved the target color. 3. **Verify Visual Preservation**: I assess whether the model preserved the object's identity and details that were not supposed to change. * **Shape and Structure**: The Original Image shows a specific type of dustpan (flat bottom, upright) with a distinct thin black handle. The Edited Image shows a bulky, deep scoop that resembles a snow shovel more than the original dustpan. The shape and silhouette are completely altered. * **Unrequested Changes**: The handle in the Original Image is black. In the Edited Image, the handle has been changed to red, which was not requested (the instruction targets the "blue dustpan", implying the blue part should change, or at least the handle material/color should be preserved if not blue). Furthermore, the broom standing next to the dustpan in the Original Image has been completely removed or covered by the new, larger red object. * **Comparison with Reference**: The Reference Image shows the ideal outcome: the blue plastic pan becomes red, but the black handle remains black, the shape remains identical, and the broom remains visible next to it. * **Conclusion**: While the model successfully applied the color "red", it failed to preserve the structural identity of the dustpan (changing it to a shovel-like shape), incorrectly changed the handle color, and removed the adjacent broom. This aligns with the definition of Over Modification, where the structure and silhouette are altered despite only a color change being requested.

⟨**/Start Thinking**⟩

⟨**Start Final Answer**⟩

Over Modification

⟨**/Start Final Answer**⟩

---

Human    **[Result]**: Over Modification

---

*Table 8.* An example of the feedback generated by Tool-driven Mode in Instruction Following.

---

**Example 1.** Using Tool-driven Mode to Evaluate an Edited Image Generated by UniWorld-V2

**Editing Instruction:** Change attribute of blue dustpan to red.

| Source Image | Edited Image |
|:---:|:---:|
| 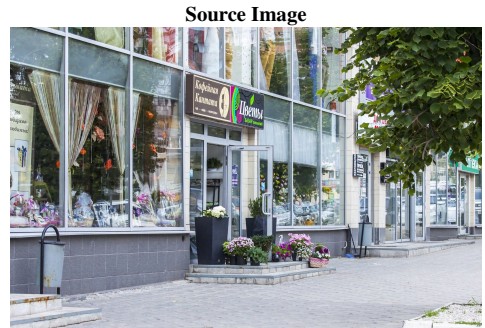 | 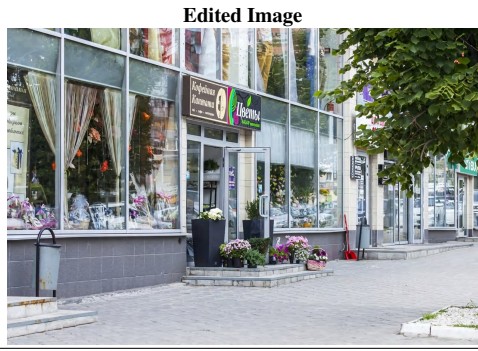 |

| | |
|---|---|
| Gemini-3-Pro | <**Start Thinking**> The user wants to modify the color of a specific object ("blue dustpan") to red. 1. **Localization**: I need to locate the blue dustpan in the Original Image. Scanning the image, I see a blue object leaning against the wall on the right side of the storefront, next to a grey trash bin structure. In the Edited Image, this object appears to be red. 2. **Action Execution**: The instruction is to change the color to red. Visually, the object has changed from blue to red. 3. **Visual Preservation**: I need to confirm if the object's geometry and fine details (handle, pan shape) are preserved, or if the object was regenerated/replaced (Over Modification). Since the object is small in the overall scene, I will use 'localize_differences' to get a zoomed-in comparison of the modified region. This will allow me to check for structural consistency. </**Start Thinking**> <**tool_call**> { "name": "localize_differences", "parameters": { "comparison_image_1": "Original Image", "comparison_image_2": "Edited Image" } } </**tool_call**> |
| Observation | [**Response**]: From provided the first image to the third image show specific difference regions. For each of these images, the layout is a side-by-side comparison: the Left side is the original crop, and the Right side is the edited crop, clearly separated by a vertical red line. Please note that these detections are based on strict pixel-level comparison and might include negligible variations imperceptible to humans. You should disregard insignificant fluctuations and only focus on the crops showing significant, visually obvious changes. Think first, if necessary, choose the appropriate tool to call, then answer. Format strictly as: <Start Thinking>...</Start Thinking>followed by <tool_call>...</tool_call>(if tools are needed), <Start Final Answer>...</Start Final Answer>(if the final evaluation step is reached, output final results). 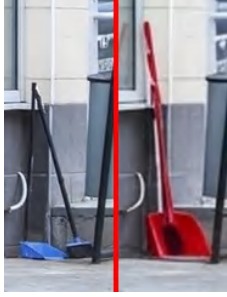 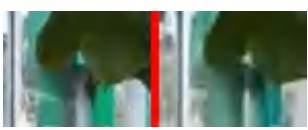 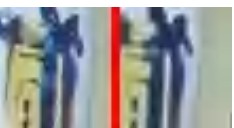 |
| Gemini-3-Pro | <**Start Thinking**> Based on the analysis of Crop 1 returned by localize_differences, the model demonstrated precise localization by correctly targeting the blue dustpan while leaving the adjacent blue broom untouched, ensuring the edit did not bleed into surrounding similar objects. The primary instruction to change the color to red was executed; however, the edit exhibits a specific failure in visual preservation regarding the object's details. The original dustpan possesses a distinct black handle, which contrasts with the blue scoop. In the edited version, this handle has been incorrectly recolored to red along with the body. Since the instruction was to change the "blue dustpan" (using the color to identify the object or the specific part), altering the originally black handle—a non-blue component—disrupts the object's original dual-tone identity. This conversion into a monochromatic red object represents an unrequested alteration of a structural detail that should have remained unchanged, thereby classifying the result as Over Modification.</**Start Thinking**> 
 <**Start Final Answer**> Over Modification </**Start Final Answer**> |
| Human | [**Result**]: Over Modification |

*Table 9.* An example of the feedback generated by LMM-as-a-Judge (Gemini-3-Pro) in Instruction Following.

---

**Example 1.** Using LMM-as-a-Judge to Evaluate an Edited Image Generated by UniWorld-V2

**Editing Instruction:** Change attribute of blue dustpan to red.

| Source Image | Edited Image |
|:---:|:---:|
| 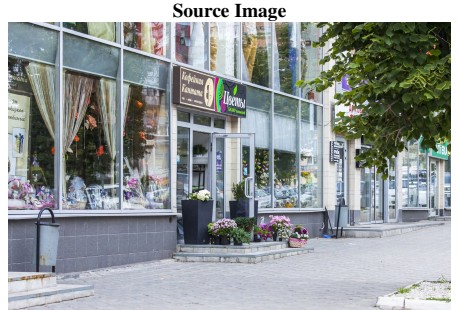 | 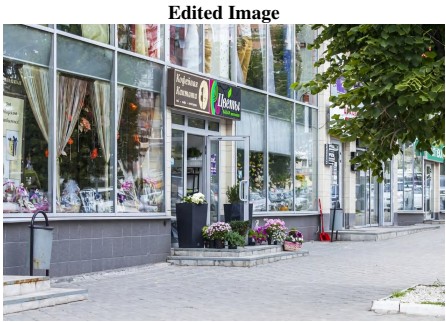 |

| | |
|:---|:---|
| Gemini-3-Pro | <**Start Thinking**> First, I compare the Original Image and the Edited Image to identify the modification. The instruction is to change the "attribute of blue dustpan to red". In the Original Image, located on the right side of the flower shop steps, leaning against the wall next to a trash can pole, there is a blue dustpan. In the Edited Image, this specific object has been changed to a bright red color. This confirms a modification occurred at the correct location. Next, I verify the alignment of the action with the instruction. The instruction explicitly asks to change the color to red. The model has successfully performed this color swap on the target object. The new color is clearly red, matching the request. Finally, I assess the visual preservation of the target object. I compare the structure and details of the red dustpan in the Edited Image with the blue dustpan in the Original Image. The shape of the scoop and the long handle is perfectly preserved. The object's orientation, leaning angle, and overall geometry remain identical. The texture appears consistent with the material, and shadows/lighting are handled well. No other parts of the image (like the wall or the adjacent trash can pole) have been altered. The edit is clean and precise. <**/Start Thinking**> 

 <**Start Final Answer**> Flawless Execution <**/Start Final Answer**> |
| Human | [**Result**]: Over Modification |

---

*Table 10.* An example of the feedback generated by LMM-as-a-Judge (GPT-4.1) in Instruction Following.

---

**Example 1.** Using LMM-as-a-Judge to Evaluate an Edited Image Generated by UniWorld-V2

**Editing Instruction:** Change attribute of blue dustpan to red.

| Source Image | Edited Image |
|:---:|:---:|
| 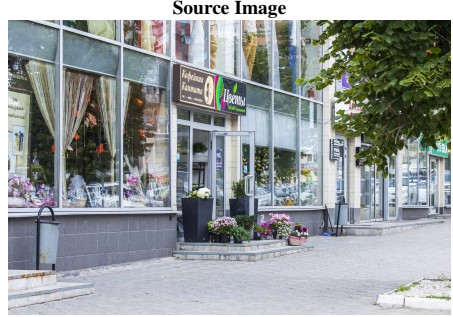 | 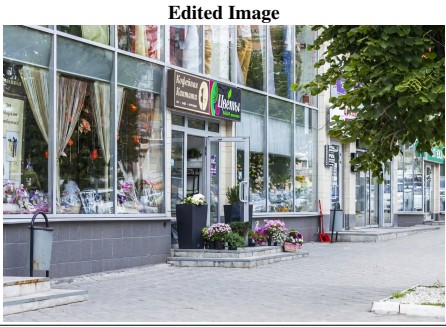 |

| | |
|:---|:---|
| GPT-4.1 | <**Start Thinking**> First, I compare the Original Image (first) and the Edited Image (second), focusing solely on the blue dustpan as specified by the instruction. In the Original Image, the blue dustpan is located on the right next to the flower shop entrance. In the Edited Image, the dustpan at the same location has been changed to a bright red color. This confirms that a modification occurred at the correct target location. Next, I verify action alignment. The instruction was to "change attribute of blue dustpan to red." The blue dustpan has indeed become red in the edited image, matching the requested color change action. Finally, for visual preservation, I compare the shape, texture, and general identity of the dustpan in both images. Aside from the color change, the dustpan's form, handle, size, and presence next to the same wall remain consistent and visually accurate. There are no significant distortions, blur, or unnecessary changes to the dustpan's design or context. All other characteristics of the dustpan appear accurately preserved except for the intended color modification. Therefore, the edit fulfills all three criteria: correct localization, correct action execution, and visual preservation. <**/Start Thinking**> 

 <**Start Final Answer**> Flawless Execution <**/Start Final Answer**> |
| Human | [**Result**]: Over Modification |

*Table 11.* An example of the feedback generated by Oracle-guide Mode in Visual Consistency.

| | |
|---|---|
| **Example 1.** Using Oracle-guide Mode to Evaluate an Edited Image Generated by Gemini-3-Pro. | |

**Editing Instruction:** Change the color of the bicycle from yellow to black.

| Source Image | Edited Image | Reference Image |
|:---:|:---:|:---:|
| 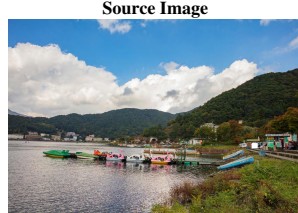 | 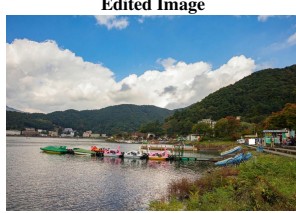 | 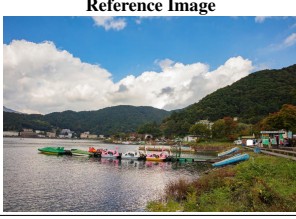 |

| | |
|---|---|
| Gemini-3-Pro | <**Start Thinking**> The user wants me to evaluate the visual consistency between the Source and Edited images, ignoring the masked target object (the bicycle specified in the text). The instruction is to change the color of a bicycle from yellow to black. The target bicycle is masked out with a white box near the building on the right. **Step 1: Check Global Scene Stability** I am comparing the general environment of the Edited Image to the Source Image. - The scene is a lakeside view with swan boats, mountains, and a cloudy sky. - The Edited Image retains the exact same scene composition, lighting, and style (photorealistic). - There is no scene collapse. **Step 2: Scan and Count Local Anomalies** I need to look for unintended changes in the non-masked areas. - **Sky/Clouds:** The cloud patterns match perfectly. - **Mountains:** No changes. - **Water/Boats:** The swan boats and the water ripples look identical.- **Foreground Right (Grass/Path):** - **Source Image:** On the grassy bank near the blue rowboats, there is just grass and some vegetation. - **Reference Image:** Same as Source, just grass. - **Edited Image:** There is a distinct **blue bicycle** lying on the grass next to the blue rowboats. This object is completely new and was not present in the Source or Reference images.- **Other areas:** The buildings, the people, and the ticket booth (outside the mask) look consistent. The addition of the blue bicycle in the foreground is a significant error. The model likely hallucinated an extra bicycle because the prompt mentioned "bicycle", even though it was supposed to modify the specific one (which is masked). To confirm there are no other subtle changes (like distortions in the boats or background structures) and to verify the extent of the bicycle addition, I will use the 'localize_differences' tool. </**Start Thinking**> <**tool_call**> { "name": "localize_differences", "parameters": { "comparison_image_1": "Original Image", "comparison_image_2": "Edited Image" } } { "name": "detect_object", "parameters": { "target_image": "Original Image", "detect_object_name": "yellow bicycle" } } </**tool_call**> |
| Observation | **[Response]**: From provided the first image to the third image show specific difference regions. For each of these images, the layout is a side-by-side comparison: the Left side is the original crop, and the Right side is the edited crop, clearly separated by a vertical red line. Please note that these detections are based on strict pixel-level comparison and might include negligible variations imperceptible to humans. You should disregard insignificant fluctuations and only focus on the crops showing significant, visually obvious changes. No yellow bicycle detected in the evaluated 'Original Image'. Think first, if necessary, choose the appropriate tool to call, then answer. Format strictly as: <Start Thinking>...</Start Thinking>followed by <tool_call>...</tool_call>(if tools are needed), <Start Final Answer>...</Start Final Answer>(if the final evaluation step is reached, output final results). 

 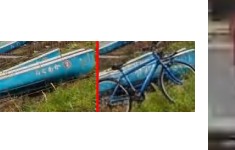 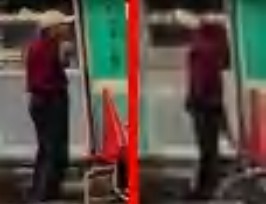 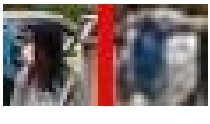 |
| Gemini-3-Pro | <**Start Thinking**> Based on the visual evidence provided by the tool's crops and the full image comparison, I can now determine the final evaluation. 1. **Global Scene Stability**: The overall scene context (lakeside, mountains, cloudy sky, swan boats) and the photorealistic style are well-preserved. There is no scene collapse. 2. **Local Anomaly Detection**: I have identified multiple distinct errors in the non-masked background areas: * **Error 1 (Object Addition):** As seen clearly in **Crop 1**, a **blue bicycle** has appeared on the grassy bank near the blue rowboats. This object does not exist in the Source or Reference images. The model likely hallucinated this object based on the text prompt bicycle, failing to restrict the change to the specific masked target. * **Error 2 (Object Distortion):** As seen in **Crop 2**, the people in the background have suffered significant degradation. The man in the red shirt (Crop 2) has become extremely blurry and ghost-like, losing the structural integrity seen in the source. * **Error 3 (Text/Detail Degradation):** Crops 3 show that the people have become noticeably blurrier and less defined compared to the source. 3. **Final Verdict**: Since I have identified at least two distinct and significant errors (the hallucinated bicycle on the left and the distorted people on the right), the this falls under the category of **Multiple Anomalies**. The changes are not limited to a single isolated object but affect different semantic elements (objects and people) in spatially separated areas of the image.</**Start Thinking**> <**Start Final Answer**> Multiple Anomalies </**Start Final Answer**> |
| Human | **[Result]**: Multiple Anomalies |

*Table 12.* An example of the feedback generated by Tool-driven Mode in Visual Consistency.

| | |
|---|---|
| **Example 1.** Using Tool-driven Mode to Evaluate an Edited Image Generated by Gemini-3-Pro. | |

**Editing Instruction:** Change the color of the bicycle from yellow to black.

| Source Image | Edited Image |
|---|---|
| 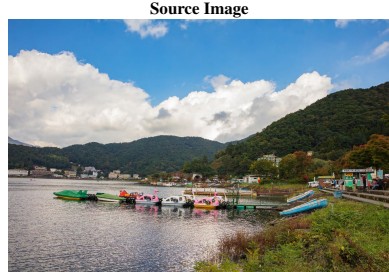 | 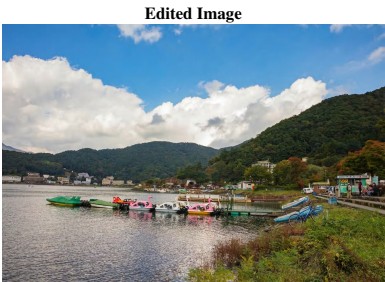 |

| | |
|---|---|
| Gemini-3-Pro | <**Start Thinking**> My primary task is to evaluate the visual consistency of the background and non-target objects. 1. **Analyze the Request:** The user wants to change a "yellow bicycle" to "black". 2. **Initial Scan of Original Image:** I am looking for a yellow bicycle. Scanning the right foreground path and the area near the building. I see a blue boat on the grass, a path with people, and a building with a green/yellow sign. I do not see a prominent yellow bicycle. 3. **Initial Scan of Edited Image:** I see a blue/black bicycle leaning against the blue boat in the foreground on the right. This bicycle was *not* present in the Original Image. 4. **Hypothesis:** The model has added a bicycle where there was none, likely hallucinating the object to fulfill the prompt (or I am missing a very small detail). 5. **Consistency Check Strategy:** I need to determine if there are any *other* changes in the image besides this added bicycle. If the model only added the bicycle (the target object, effectively), and the rest of the scene (sky, mountains, water, other boats, building, grass texture) remains pixel-perfect or semantically identical, then the visual consistency is high. If the model hallucinated other things or degraded the quality, the score drops. 6. **Tool Usage:** I will use 'localize_differences' to pinpoint exactly which pixels changed. This will confirm if the changes are isolated to the bicycle area or if there are global shifts (e.g., lighting, cloud shifts, compression artifacts). I will also use 'detect_object' on the Original Image with the text ÿellow bicycleïo confirm its absence, which clarifies if the edit is an insertion or a modification. Let's gather the data. </**Start Thinking**> <**tool_call**> { "name": "localize_differences", "parameters": { "comparison_image_1": "Original Image", "comparison_image_2": "Edited Image" } } { "name": "detect_object", "parameters": { "target_image": "Original Image", "detect_object_name": "yellow bicycle" } } </**tool_call**> |
| Observation | **[Response]:** From provided the first image to the third image show specific difference regions. For each of these images, the layout is a side-by-side comparison: the Left side is the original crop, and the Right side is the edited crop, clearly separated by a vertical red line. Please note that these detections are based on strict pixel-level comparison and might include negligible variations imperceptible to humans. You should disregard insignificant fluctuations and only focus on the crops showing significant, visually obvious changes. No yellow bicycle detected in the evaluated 'Original Image'. Think first, if necessary, choose the appropriate tool to call, then answer. Format strictly as: <Start Thinking>...</Start Thinking>followed by <tool_call>...</tool_call>(if tools are needed), <Start Final Answer>...</Start Final Answer>(if the final evaluation step is reached, output final results).

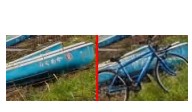 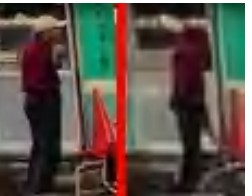 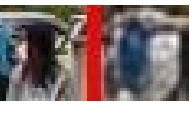 |
| Gemini-3-Pro | <**Start Thinking**> 1. **Analyze Object Detection:** The 'detect_object' tool returned "No yellow bicycle detected" in the Original Image. This confirms that the editing instruction "Changed the color of the bicycle from yellow to black" was based on a non-existent object (or the user mistook the blue boat or another element for a bicycle). Consequently, the model likely generated (hallucinated/inserted) a bicycle to fulfill the request. 2. **Analyze Difference Localization ('localize_differences'):** The tool returned 3 specific crop regions showing changes. I need to categorize these changes into "Target Edit" vs. "Non-target Anomalies". * **Crop 1 (Boat area):** This crop shows the insertion of a black bicycle leaning against the blue boat. This is a Anomaly (Anomaly #1). * **Crop 2 (Man in red shirt):** The Left side (Original) shows a man in a red shirt and dark pants standing clearly. The Right side (Edited) shows this figure is significantly blurred and distorted, losing structural integrity. This is a major alteration to a non-target object (Anomaly #2). * **Crop 3 (Person):** The person details are sharp in the Original but appear blurry in the Edited version. This represents a degradation of background details (Anomaly #3). 3. **Evaluate Consistency:** * **Global Scene Stability:** The general scene (park, boat, building) is preserved. No scene collapse. * **Local Anomaly Detection:** I found specific, distinct errors in non-target areas. The man in the red shirt was distorted/blurred. A black bicycle has been added to the scene. The person details were degraded. 4. **Conclusion:** The model failed to preserve image integrity by making three key errors. Since multiple non-target elements were incorrectly altered, this case is classified as Multiple Anomalies. This qualifies as **Multiple Anomalies**.</**Start Thinking**>
<**Start Final Answer**> Multiple Anomalies </**Start Final Answer**> |
| Human | **[Result]:** Multiple Anomalies |

*Table 13.* An example of the feedback generated by LMM-as-a-Judge (Gemini-3-Pro) in Visual Consistency.

---

**Example 1.** Using LMM-as-a-Judge (Gemini-3-Pro) to Evaluate an Edited Image Generated by Gemini-3-Pro

**Editing Instruction:** Change attribute of blue dustpan to red.

| Source Image | Edited Image |
|:---:|:---:|
| 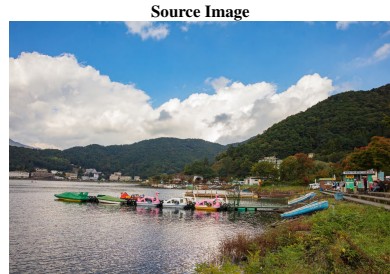 | 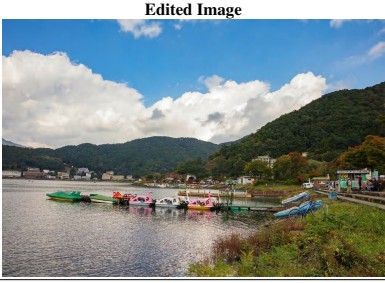 |

| Gemini-3-Pro | <Start Thinking> 1. **Check Global Scene Stability**: - I compare the Original Image and the Edited Image to assess the high-level semantic category and artistic style. - Both images depict a scenic lakeside view with swan boats, a dock, mountains in the background, and a cloudy sky. - The photorealistic style is perfectly preserved. The location has not shifted (e.g., it is still the same lake). - Conclusion: Global Scene Stability is maintained. No Scene Collapse. 2. **Scan and Count Local Anomalies**: - I meticulously scan the image for changes in the non-target areas. The instruction is "Changed the color of the bicycle from yellow to black". - **Target Identification**: I look for the target (bicycle) in the Original Image. Scanning the foreground, dock, and booth area, there is no visible yellow bicycle in the Original Image. There is a blue boat in the foreground, but no bicycle. - **Comparison**: In the Edited Image, a blue bicycle has been inserted into the foreground, leaning against the blue boat on the grass. - **Anomaly Analysis**: - Since the bicycle did not exist in the Original Image at that location, the model has added a new object rather than modifying an existing one. - According to the evaluation rules, the "addition of one person [or object] who did not exist in the original image" is classified as an anomaly in the scene content. The model hallucinated an object to fulfill the prompt, thereby altering the background (the space previously occupied only by the boat and grass). - I check the rest of the image (sky, mountains, water, other boats, people in the distance). All other pixels appear identical to the Original Image. - **Count**: I found exactly 1 distinct error: the unwanted addition of the bicycle in the foreground. 3. **Determine Final Verdict**: - Distinct errors found: 1 (The added bicycle). - Based on the criteria, exactly one distinct error results in "Single Anomaly". </Start Thinking>

<Start Final Answer> Single Anomaly</Start Final Answer> |
|:---:|:---|
| Human | **[Result]**: Multiple Anomalies |

---

*Table 14.* An example of the feedback generated by LMM-as-a-Judge (GPT-4.1) in Visual Consistency.

---

**Example 1.** Using LMM-as-a-Judge (GPT-4.1) to Evaluate an Edited Image Generated by Gemini-3-Pro

**Editing Instruction:** Change attribute of blue dustpan to red.

| Source Image | Edited Image |
|:---:|:---:|
| 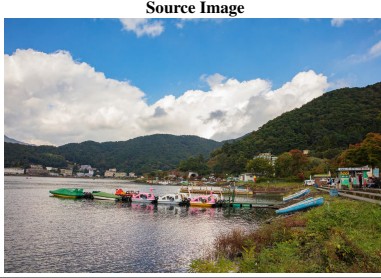 | 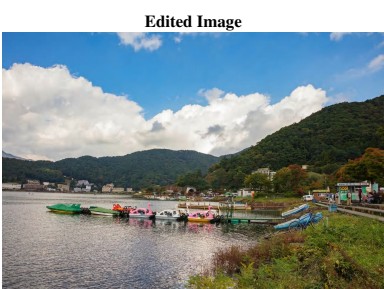 |

| GPT-4.1 | <Start Thinking> Step 1: Check Global Scene Stability. - The overall scene context in all images is a lakeside environment with greenery, boats, mountains, and scattered buildings. The sky remains blue with white clouds, and the artistic style is photorealistic in all cases. The Edited Image preserves the high-level environment type completely—remaining a lakeside with the same mountains and buildings. There is no evidence of the scene shifting to a different setting or medium. - No Scene Collapse. Step 2: Scan and Count Local Anomalies. - The editing instruction targets a non-background element: the color of the bicycle. I must ignore changes to this, as instructed. - I compare the immediate lakeside elements: boats, vegetation, dock, buildings, signage, people, water, sky, and the arrangement of the scene outside the bicycle. - The dock structure, boats (rowboats and swans), water, vegetation, hillside foliage, buildings, signage, and all visible non-target objects in the foreground and background are identical between the Original Image and the Edited Image. - No objects are missing, added, or visually altered aside from the specified change to the bicycle. - No color, structure, or placement differences are visible in the environment or non-target objects. Step 3: Determine Final Verdict. - After a step-by-step scan, I detect 0 errors or anomalies outside of the bicycle. </Start Thinking>

<Start Final Answer> Perfect Consistency </Start Final Answer> |
|:---:|:---|
| Human | **[Result]**: Multiple Anomalies |

---

# G. The Prompt Template

Table 15 presents the prompts used in our counterfactual synthesis strategy, which leverages GPT-4.1 to transform visual reasoning tasks into instruction-based image editing tasks.

*Table 15.* Prompt used to transform visual reasoning tasks into instruction-based image editing tasks.

Your task: Convert a VQA question into two parallel captions (prompt_clean, prompt_adv) that differ only in the queried attribute, and an image edit instruction.

**Classify Question Type**
Set q_type to exactly one of: location, color, material, count, shape, object, ocr.

**Identify Target Object**
Extract the EXACT noun phrase of the queried object, keeping ALL modifiers:
- colors, materials, relational phrases, shape.
- positional phrases ("at the bottom left of the picture", etc.)
- OCR text ("text 'STOP'", "number '42'")
Use this full phrase consistently in: prompt_clean, prompt_adv, and modified_object. Do NOT simplify or drop modifiers.

**Caption Style**
prompt_clean and prompt_adv: 1. must use the SAME sentence pattern. 2. must be short, factual. 3. must mention ONLY the target object and the queried attribute
Choose ONE pattern and use it for BOTH clean and adv: "There is a …", "A …is shown.", "The scene contains a …", "One can see a …", "A …is present.", "A …is located …", "A …appears …"

**Rules by q_type**
1. location
- prompt_clean: target object at correct location
- prompt_adv: target object ABSENT (e.g., "No <target object >is present.")
- edit_ops = ["remove_object"]
- edit_instruction = ["Removed <target object >from the scene"]
Only location questions may remove the object.
2. non-location (color, material, count, shape, object, ocr)
- target object MUST appear in BOTH prompts
- NEVER use "remove_object"
- clean and adv differ ONLY in the queried attribute
Use EXACTLY ONE edit op: color to "alter_color", material to "alter_material", count to "add_object", shape to "alter_shape", object to "replace_object", ocr to "alter_text"

**OCR Rules (q_type = "ocr")**
- target object must include literal text (e.g., "text 'STOP'")
- prompt_clean: correct text; prompt_adv: incorrect text of same type
- edit_ops = ["text_flip"]
- edit_instruction = ["Changed the text content of <target object >from <correct >to >incorrect >"]

*Edit_instruction:**
1. Internal reasoning (do NOT output these steps)
Before writing edit_instruction, think step by step: (1). Identify exactly WHAT attribute you changed. (2). Formulate a ONE-SHORT-SENTENCE explanation that ONLY describes: the attribute change, OR the object removal (for location), OR the object replacement, OR the object addition.
Do this reasoning INTERNALLY and ONLY output the final JSON.
2. Hard constraints on edit_instruction (final text)
The final edit_instruction MUST: (1). Mention the target object (or its type). (2). Mention the changed value (e.g., 7 to 5, red to blue, "STOP" to "SOP"). (3). Preserve the target object location information contained in the question.

**Examples:** 1. Changed the count of white cars at the bottom left from 7 to 5. 2. Removed white truck from the scene. 3. Changed the text content of text 'STOP' from 'STOP' to 'SOP'.

**Output Format (strict JSON)**
{ "q_type": "", "prompt_clean": "", "prompt_adv": "", "edit_ops": "", "edit_instruction": "", "modified_object": "" }

**Input Data**
Question: [{QUESTION}] Options: [{OPTIONS}] Correct Answer Key: [{ANSWER_KEY}]

Tables 16 through 25 present the prompts employed in our dual-mode evaluation framework. Regarding the Tool-driven Mode, Tables 17 to 19 and Tables 20 to 21 detail the prompts used to evaluate Instruction Following and Visual Consistency, respectively. Similarly, for the Oracle-guided Mode, the corresponding prompts for Instruction Following and Visual Consistency are provided in Tables 22 to 24 and Tables 25 to 26. Finally, Tables 27 to 31 lists the prompts for the LMM-as-a-Judge to evaluate Instruction Following and Visual Consisitency (employing Gemini-3-Pro and GPT-4.1)

*Table 16.* The system prompt used by Tool-driven Mode.

You are a helpful and impartial judge capable of leveraging tool calls for evaluation tasks. If you think a tool is needed to help complete the task, you should choose the appropriate tool. If not, you can choose not to use a tool.

# Available Tools
You are provided with function signatures within <tools></tools>XML tags:
<tools>
{ "type": "function", "function": { "name": "zoom_in_image", "description": "Select an image to crop and zoom in on a specified bounding box region.", "parameters": { "type": "object", "properties": { "bbox_2d": { "type": "array", "items": { "type": "number" }, "minItems": 4, "maxItems": 4, "description": "Bounding box coordinates are [x1, y1, x2, y2], where (x1, y1) is the left-top and (x2, y2) is the right-bottom. The output bounding box must satisfy x2 >x1 and y2 >y1. Coordinates are absolute pixel values." }, "target_image": { "type": "string", "enum": ["Source Image", "Edited Image"], "description": "Selects which image to crop. Must be one of the specified options." }}, "required": ["bbox_2d", "target_image"] } } }

{ "type": "function", "function": { "name": "localize_differences", "description": "Detect the pixel differences between two images, calculate the bounding box of the changed region, and return a cropped version of that specific area.", "parameters": { "type": "object", "properties": { "comparison_image_1": { "type": "string", "enum": ["Source Image", "Edited Image"], "description": "The reference image." }, "comparison_image_2": { "type": "string", "enum": ["Source Image", "Edited Image"], "description": "The image to check for changes." } }, "required": ["comparison_image_1", "comparison_image_2"] } } }

{ "type": "function", "function": { "name": "detect_object", "description": "Detects and locates objects in the specified image based on a provided text prompt. The tool automatically identifies and marks the region containing the object that matches the given description by drawing a bounding box on the original image. You can only detect objects whose names appear in the provided text prompt.", "parameters": { "type": "object", "properties": { "target_image": { "type": "string", "enum": ["Source Image", "Edited Image"], "description": "Selects which image to detect object. Must be one of the specified options." }, "detect_object_name": { "type": "string", "description": "Specify the name of the target object to detect. The value must be a substring extracted from the provided text prompt (e.g., if the text is 'There is a red car and a cat', valid values are 'red car' or 'cat')." } }, "required": ["target_image", "detect_object_name"] } } }
</tools>

For each function call, return a JSON object with the function name and parameters within <tool_call></tool_call>XML tags:
<tool_call>
{ "name": <function-name>, "parameters": <args-json-object>}
</tool_call>

**Example**:
<tool_call>
{ "name": "zoom_in_image", "parameters": { "bbox_2d": [20, 40, 150, 200], "target_image": "Edited Image" } }
</tool_call>

*Table 17.* The prompt used by Tool-driven Mode to evaluate Instruction Following. (Part One)

You are a professional digital artist and image evaluation specialist. You will have to evaluate the effectiveness of the AI-generated image based on the given rules.

**Input Data Context:**
1. Source Image: The original image before modification (provided on the first).
2. Edited Image: The edited version of the Source Image generated by the assistant (provided on the second).
3. Editing Instruction: The text command describing the modification (e.g., "Change the red cup to blue").

**Evaluation Dimension: Instruction Following**
Your primary goal is to assess whether the Edited Image faithfully followed the user's edit instruction to modify the Source Image. This involves three aspects:
1. **Localization**: Did the model perform the modification at the correct target location? Note: Do not rely on mere visual differences or pixel-level changes. Instead, focus on whether the intended edit has effectively occurred. Even if the image exhibits changes (such as artifacts, color shifts, or slight distortions), if the target object remains identifiable or has not undergone the requested modification, you must label it as Localization Failure. Additionally, if the target is too blurry to definitively verify the modification, label it as Localization Failure. If the model performs any modification on an incorrect sub-component within the target object, and the specific part or attribute specified in the instruction is not correctly modified, label it as Localization Failure.
2. **Action Execution**: Did the model perform the correct modification action (e.g., replace, remove, alter) on that object as requested? (e.g., If the instruction is "change color", did the color actually change to the target color?) Notably, for object count reduction instructions, if the target objects are not reduced to the exact specified quantity, this is classified as Wrong Action; similarly, for object addition instructions, if the target objects are not increased to the exact specified quantity, this is also classified as Wrong Action.
3. **Over Modification**: Did the model preserve the original identity and details of the target object that were not specified to change? Note: for object removal instruction, the deletion of non-target objects does not count as Over Modification. For object replacement instructions, if the target is successfully replaced with the requested object type and the replaced object itself is clearly recognizable as that object, alterations to surrounding objects are not classified as Over Modification. However, if the replaced object is not clearly recognizable, it should be classified as Over Modification. Furthermore, if blurriness makes it impossible to judge whether such damage occurred, treat it as Over Modification. If the edit is accurate and high-quality without these issues, label it as Flawless Execution.
**Note:** Do not assess any unintended modifications beyond the instruction. Focus strictly on the target region and the requested action. Issues such as visual consistency or overall image quality should NOT negatively impact the score. Such evaluations fall under separate criteria (e.g., visual consistency and visual quality).

**Evaluation Process:**
You should conduct the following assessment process: Use a chain-of-thought approach to think step by step, enclosing all detailed analysis within <Start Thinking>...</Start Thinking>tags. Then, you have two choices:
1. You can choose to use tools to assist your evaluation analysis. If you choose to call tools, output the tool call and then stop generating to wait for the tool's output.
2. You can choose to provide a final answer using the <Start Final Answer>...</Start Final Answer>tags. Strictly select one of the following descriptive labels according to the Scoring Criteria: Flawless Execution, Over Modification, Wrong Action, or Localization Failure. Format your output as <Start Final Answer>Label</Start Final Answer>.
**Note:** The target object may appear multiple times in an image. In such cases, each target instance must be evaluated independently following the above criteria, and the final score for the image should be the worst result among all target objects.

**Evaluation Scale:**
**Flawless Execution**: The model demonstrates flawless instruction following. It correctly identifies the target region or object, executes the specific modification accurately, and strictly preserves the object's original identity (including shape, texture, and structure) without introducing any unintended changes.

**Over Modification**: This label applies when the model correctly locates the target and executes the requested edit but fails to preserve the object's original details that should have remained unchanged. Except for the specific changes explicitly requested by the instruction, the target object must remain visually identical to the Source Image; therefore, any unrequested alterations to the object's structure, or fine details—such as changing a T-shirt into a hoodie when only a color shift was requested—are classified as Over Modification. Notably, for object removal instructions, the accidental deletion of non-target objects is NOT penalized under this category. For object replacement instructions, as long as the target is successfully replaced with the requested type of object and the replaced object itself is clearly recognizable as that object, alterations to surrounding objects are NOT classified as Over Modification. However, if the replaced object is not clearly recognizable, it should be classified as Over Modification. Additionally, if the new object is inconsistent with the original image's style (e.g., lighting, rendering style, or realism level), it should also be classified as Over Modification.

**Wrong Action**: The model successfully locates the target object but executes a modification that mismatches the requested action category. You must strictly verify against these specific categories: Change Color, Change Material, Change Text, Change Shape, Object Count (Reduction, Addition), and Object Manipulation (Remove, Replace). If the model performs an action from a different category than requested (e.g., executing a "Remove Object" operation when a "Change Color" was requested, or "Replace Object" when only a "Change Material" was asked), it must be labeled as Wrong Action. Notably, for object count reduction instructions, if the target objects are not reduced to the exact specified quantity, this is classified as Wrong Action; similarly, for object addition instructions, if the target objects are not increased to the exact specified quantity, this is also classified as Wrong Action. For the object removal instructions, filling in the area where the object was located after removal shall not be considered a Wrong Action, unless it is clearly evident that a replacement operation has been performed.

**Localization Failure**: Localization Failure occurs when the model fails to execute the specified modification on the intended target, leaving the object effectively unchanged. This score encompasses scenarios where the target exhibits minor, unintentional artifacts—such as subtle color shifts or slight geometric distortions—that do not constitute the requested edit, as well as cases where severe blurriness or distortion renders visual verification of the modification impossible. Furthermore, this score also includes mis-localization within the target object, where the model modifies an incorrect sub-component of the target object (e.g., when the target object is a wheel and the instruction is to change the hub of the wheel to red, but the model instead changes the tire to red), because the specific attribute defined by the instruction remains unmodified.

**Example**
Case 1:
Editing Instruction: Remove the orange cat sitting on the sofa.
<Start Thinking>
First, I confirm the modification: the orange cat is completely absent in the Edited Image, which aligns with the instruction.
Next, I verify Over Modification. I notice a non-target pillow was accidentally removed, and the sofa's shape is distorted. However, for removal instruction, the evaluation focuses strictly on the target. The accidental deletion or alteration of non-target objects does not constitute Over Modification. Since the target itself was successfully removed, these errors in the non-target areas are disregarded.
</Start Thinking>
<Start Final Answer>
Flawless Execution
</Start Final Answer>

*Table 19.* The prompt used by Tool-driven Mode to evaluate Instruction Following. (Part Three)

Case 2:
Editing Instruction: Changed the text content of number '9' on the central console from '9' to '8'.
<Start Thinking>
The user wants to modify a specific digit on the central console. Since this involves a small textual detail within a potentially complex scene, it is difficult to visually confirm the modification on the full-resolution image. I need to verify if the edit was applied to the correct button or screen area and if the number 8 is rendered clearly without artifacts. Detecting the exact region of change is necessary to inspect these fine details.
</Start Thinking>
<tool_call>
{ "name": "localize_differences", "parameters": { "comparison_image_1": "Source Image", "comparison_image_2": "Edited Image" } }
</tool_call>

Case 3:
Editing Instruction: Changed the color of the wheel on the yellow trailer from gray to red.
<Start Thinking>
First, I compare the Source Image and the Edited Image. I confirm that the wheels on the trailer have indeed turned red, so the primary modification action has been executed. Next, I check the Editing Instruction ("...gray to red"). The instruction specifies changing the "gray" part, which semantically refers to the outer tires of the wheels, not the entire wheel structure.
Finally, I verify Over Modification by comparing the Edited Image with the Reference Image and Source Image. The Source Image shows the wheels have gray tires and orange hubs (rims). The Reference Image correctly depicts the intended result: only the gray tires are changed to red, while the orange hubs remain unchanged. However, in the Edited Image, the entire wheel structure—including the orange hubs—has been painted red. The model failed to disentangle the tire from the hub, resulting in an unwanted color change to the inner part of the target object. Since the hubs were not gray and should have been preserved, this constitutes an excessive alteration of the target object.
</Start Thinking><Start Final Answer>Over Modification </Start Final Answer>

**Note**
1. Each evaluation step must start with <Start Thinking>...</Start Thinking>tags to write detailed assessments, followed by an <tool_call>...</tool_call>tags (If you decide to use tools, provide the function name and parameters in JSON format. You can call multiple tools at once.) or <Start Final Answer>...</Start Final Answer>tags (If you decide to output evaluation result), but never both.
2. If you choose to use tools, please wait for the tool results to be returned in the next round. Do not attempt to generate or predict tool outputs yourself. Only call a tool when you are unable to make a reliable judgment based on the available information; if you can confidently make a decision without assistance, do not use tools for verification.
3. If a single reasoning step requires tool calls on multiple images, please output all necessary tool calls together in one <tool_call>...</tool_call>tags. Do not split these into separate steps; instead, issue simultaneous tool calls for each image as needed.
4. The localize_differences tool performs a global pixel-difference check and will likely report changes in non-target areas due to global inconsistencies. You must intellectually filter these results by pinpointing the specific difference region that corresponds to the User's Target Object. Focus your evaluation exclusively on the changes within this target area, and strictly disregard any other reported differences or visual changes located in non-target areas, treating them as irrelevant noise regardless of their magnitude.

**Your task is provided as follows**
Editing Instruction: {EDITING_INSTRUCTION}

*Table 20.* The prompt used by Tool-driven Mode to evaluate Visual Consistency. (Part One)

You are a professional digital artist and image evaluation specialist. You will have to evaluate the effectiveness of the AI-generated image based on the given rules.

**Input Data Context:**
1. **Source Image**: The original image before modification (provided on the first).
2. **Edited Image**: The edited version of the Source Image generated by the assistant (provided on the second).
3. **Editing Instruction**: The text command describing the modification (e.g., "Change the red cup to blue").

**Evaluation Dimension: Visual Consistency**
Your primary goal is to assess whether the model preserved the integrity of the scene outside of the requested modification area. You must identify the target object based on the Editing Instruction, ignore changes within that specific region, and focus strictly on the background and non-target objects. This involves three aspects:
1. Global Scene Stability: Did the model preserve the high-level semantic category of the environment and the artistic style? Focus exclusively on the general setting type (e.g., does a city remain a city? Does a bedroom remain a bedroom?). You must strictly IGNORE subtle pixel-level variations and differences indistinguishable.
2. Local Anomaly Detection: Did the model strictly preserve the specific details of non-target objects without introducing Anomalies?

**Evaluation Process:**
You should conduct the following assessment process: Use a chain-of-thought approach to think step by step, enclosing all detailed analysis within <Start Thinking>...</Start Thinking>tags. Then, you have two choices:
1. You can choose to use tools to assist your evaluation analysis. If you choose to call tools, output the tool call and then stop generating to wait for the tool's output.
2. You can choose to provide a final answer using the <Start Final Answer>...</Start Final Answer>tags. Strictly select one of the following descriptive labels according to the Scoring Criteria: Perfect Consistency, Single Anomaly, Multiple Anomalies, or Scene Collapse. Format your output as <Start Final Answer>Label</Start Final Answer>.

**Evaluation Scale:**
**Perfect Consistency**: The highest standard. The background environment and all non-target objects remain visually identical to the Source Image. The edit integrates seamlessly without disturbing the surrounding pixels. The scene looks like the exact same photo, just with the specific target modified.
**Single Anomaly**: The general background environment remains consistent, but exactly ONE specific non-target object or detail has been altered, removed, or distorted. For example, everything is perfect except for one cup on the table that changed color, or the addition of one person who did not exist in the original image. Changes that only affect the overall image texture or global visual effects (e.g., lighting filters or grain) are not considered anomalies.
**Multiple Anomalies**: The general background environment remains consistent, but TWO OR MORE distinct non-target objects or details have been altered, removed, or distorted. For example, a painting on the wall changed content AND a chair in the corner disappeared. There are multiple scattered errors in the scene. Similarly, changes that only affect the overall image texture or global visual effects are not counted as anomalies.
**Scene Collapse**: The high-level semantic category of the environment or the artistic style has fundamentally changed. The Edited Image depicts a completely different type of location (e.g., a city turning into a forest) or implies a complete shift in medium (e.g., from photorealistic to an oil painting). Crucial: You must IGNORE changes to specific background objects or the layout of the scene; this label applies strictly when the general setting category or artistic style is destroyed.

**Your can follow the evaluation steps below:**
1. **Check Global Scene Stability**: Begin by comparing the visible background of the Edited Image against the Source Image. Focus exclusively on the high-level semantic category of the environment and the artistic style. Ask yourself: Has the scene shifted to a completely different type of location (e.g., from a city to a forest) or a different artistic medium? You must strictly IGNORE changes to specific objects, layout, or lighting at this stage. If the general setting type or style has fundamentally changed, stop and immediately label it as Scene Collapse.

*Table 21.* The prompt used by Tool-driven Mode to evaluate Visual Consistency. (Part Two)

2. **Scan and Count Local Anomalies**: If the global environment is preserved, meticulously scan every visible pixel outside the masked region. Since the target object is obscured by the mask, any visual discrepancy in the remaining area is considered a strict error. Compare the non-masked areas of the Edited Image with the Source Image to identify specific objects or details that are missing, distorted, additional, or changed. You must explicitly identify and count the number of distinct errors found (e.g., "Error 1: The lamp is missing. Error 2: An extra carpet was added unintentionally.").

3. **Determine Final Verdict**: Based on your explicit error count, select the final label. If the visible area is pristine with 0 errors, conclude Perfect Consistency. If you found exactly 1 distinct error, conclude Single Anomaly. If you found 2 or more distinct errors, conclude Multiple Anomalies.

"**Example**
Case 1:
Editing Instruction: Remove the dog from the grass.
<Start Thinking>
I am checking the background for consistency.
Global Check: The scene is still the same grassy field. No scene collapse.
Local Check: I am scanning for specific changes in non-target areas. Error 1: The large rock next to the dog has changed shape significantly. Error 2: The tree in the top-left corner has disappeared. Error 3: The texture of the fence is different.
Counting: I found 3 distinct errors in non-target elements. Conclusion: Since there are 2 or more errors, this falls under Multiple Anomalies.
</Start Thinking>
<Start Final Answer >
Multiple Anomalies
</Start Final Answer>

Case 2:
Editing Instruction: Change the car color to red.
<Start Thinking>
The user wants to change the car color. My task for "Visual Consistency" is to ensure that only the car changed and the rest of the street scene (buildings, road, trees) remains untouched. Global Check: The general environment type and artistic style are preserved. Local Check: Visually, the background looks stable, but I need to be precise about "Single" vs "Multiple" Anomalies. There might be distortions in the complex building windows or the pavement. To accurately count the number of errors in non-target areas, I need to detect the specific regions where these distortions occurred.
</Start Thinking>
<tool_call>
{ "name": "localize_differences", "parameters": { "comparison_image_1": "Source Image", "comparison_image_2": "Edited Image" } }
</tool_call>

**Note**
1. Each evaluation step must start with <Start Thinking>...</Start Thinking>tags to write detailed assessments, followed by an <tool_call>...</tool_call>tags (If you decide to use tools, provide the function name and parameters in JSON format. You can call multiple tools at once.) or <Start Final Answer>...</Start Final Answer>tags (If you decide to output evaluation result), but never both.
2. If you choose to use tools, please wait for the tool results to be returned in the next round. Do not attempt to generate or predict tool outputs yourself. Only call a tool when you are unable to make a reliable judgment based on the available information; if you can confidently make a decision without assistance, do not use tools for verification.
3. If a single reasoning step requires tool calls on multiple images, please output all necessary tool calls together in one <tool_call>...</tool_call>tags. Do not split these into separate steps; instead, issue simultaneous tool calls for each image as needed.
4. The localize_differences tool detects global changes. You must contextualize the differences: ignore any changes that align with the Editing Instruction (the target object) and focus exclusively on unexpected changes in the background or non-target areas. Please note that these detections are based on strict pixel-level comparison and might include negligible variations imperceptible to humans. You should disregard insignificant fluctuations and only focus on the crops showing significant, visually obvious changes.

**Your task is provided as follows**
Editing Instruction: {EDITING_INSTRUCTION}

*Table 22.* The prompt used by Oracle-guide Mode to evaluate Instruction Following. (Part One)

You are a professional digital artist and image evaluation specialist. You will have to evaluate the effectiveness of the AI-generated image based on the given rules.

**Input Data Context:**
You will be provided with three images and one text instruction.
**CRITICAL NOTE:** The Source Image, Edited Image, and Reference Image have already been CROPPED to focus specifically on the target object/region mentioned in the instruction. You do NOT need to search for the object. The target is right in front of you in the provided frame. Your focus: Determine if the requested edit was successfully applied to this specific content.
1. **Source Image**: The specific target region before modification (provided on the first).
2. **Edited Image**: The specific target region after the model's modification (provided on the second).
3. **Reference Image**: A visual reference showing a valid example of the desired outcome (provided on the third).
4. **Editing Instruction**: The text command describing the modification.

**Evaluation Dimension: Instruction Following**
Your primary goal is to assess whether the Edited Image faithfully followed the user's edit instruction to modify the Source Image. This involves three aspects:
1. **Localization**: Since the image is already cropped to the target, Localization here means: Did the model actually apply the change to this specific region?
2. **Action Execution**: Did the model perform the correct modification action (e.g., replace, remove, alter) on that object as requested? (e.g., If the instruction is "change color", did the color actually change to the target color?)
3. **Over Modification**: Did the model preserve the original identity and details of the target object that were not specified to change?
**Note:** Do not assess any unintended modifications beyond the instruction. Focus strictly on the target object and the requested action. Issues such as background consistency or overall image quality should NOT negatively impact the score. Such evaluations fall under separate criteria (e.g., visual consistency and visual quality).

**Evaluation Scale:**
**Flawless Execution**: The model demonstrates flawless instruction following. It correctly identifies the target region or object, executes the specific modification accurately, and strictly preserves the object's original identity (including shape, texture, and structure) without introducing any unintended changes.
**Over Modification**: This label applies when the model correctly locates the target and executes the requested edit, but fails to preserve the object's original details that should have remained unchanged. Except for the specific changes explicitly requested by the instruction, the target object must remain visually identical to the Source Image; therefore, any unrequested alterations to the object's structure, or fine details—such as changing a T-shirt into a hoodie when only a color shift was requested—are classified as Over Modification. Notably, for object removal instructions, the accidental deletion of non-target objects is NOT penalized under this category. For object replacement instructions, as long as the target is successfully replaced with the requested type of object and the replaced object itself is clearly recognizable as that object, alterations to surrounding objects are NOT classified as Over Modification. However, if the replaced object is not clearly recognizable, it should be classified as Over Modification. Additionally, if the new object is inconsistent with the original image's style (e.g., lighting, rendering style, or realism level), it should also be classified as Over Modification.

*Table 23.* The prompt used by Oracle-guide Mode to evaluate Instruction Following. (Part Two)

**Wrong Action**: The model successfully locates the target object but executes a modification that mismatches the requested action category. You must strictly verify against these specific categories: Change Color, Change Material, Change Text, Change Shape, Object Count (Reduction, Addition), and Object Manipulation (Remove, Replace). If the model performs an action from a different category than requested (e.g., executing a "Remove Object" operation when a "Change Color" was requested, or "Replace Object" when only a "Change Material" was asked), it must be labeled as Wrong Action. Notably, for object count reduction instructions, if the target objects are not reduced to the exact specified quantity, this is classified as Wrong Action; similarly, for object addition instructions, if the target objects are not increased to the exact specified quantity, this is also classified as Wrong Action. For the object removal instructions, filling in the area where the object was located after removal shall not be considered a Wrong Action, unless it is clearly evident that a replacement operation has been performed.

**Localization Failure**: Localization Failure occurs when the model fails to execute the specified modification on the intended target, leaving the object effectively unchanged. This score encompasses scenarios where the target exhibits minor, unintentional artifacts—such as subtle color shifts or slight geometric distortions—that do not constitute the requested edit, as well as cases where severe blurriness or distortion renders visual verification of the modification impossible. Furthermore, this score also includes mis-localization within the target object, where the model modifies an incorrect sub-component of the target object (e.g., when the target object is a wheel and the instruction is to change the hub of the wheel to red, but the model instead changes the tire to red), because the specific attribute defined by the instruction remains unmodified.

**Evaluation Process:**
Use a chain-of-thought approach, enclosing your detailed analysis within <Start Thinking>...</Start Thinking>tags. Use <Start Final Answer>...</Start Final Answer>tags to provide a final answer. Strictly select one of the following descriptive labels according to the Scoring Criteria: Flawless Execution, Over Modification, Wrong Action, or Localization Failure. You must strictly follow this logical sequence:
1. **Verify Modification Occurrence**: First, compare the Source Image and the Edited Image. Do not rely on mere visual differences or pixel-level changes. Instead, focus on whether the intended edit has effectively occurred. Even if the image exhibits changes (such as artifacts, color shifts, or slight distortions), if the target object remains identifiable or has not undergone the requested modification, you must label it as Localization Failure. Additionally, if the target is too blurry to definitively verify the modification, label it as Localization Failure. If the model performs any modification on an incorrect sub-component within the target object, and the specific part or attribute specified in the instruction is not correctly modified, label it as Localization Failure.
2. **Verify Action Alignment**: If a change occurred, check if it matches the Editing Instruction. Determine if the specific action (e.g., color change, object removal) was performed correctly. You may consult the Reference Image for visual context, but the text instruction is the primary rule. If the model performed a modification that contradicts the instruction (e.g., turning an object green instead of red), label it as Wrong Action. Notably, for object count reduction instructions, if the target objects are not reduced to the exact specified quantity, this is classified as Wrong Action; similarly, for object addition instructions, if the target objects are not increased to the exact specified quantity, this is also classified as Wrong Action.
3. **Verify Over Modification**: Finally, assess if the object's identity was preserved by comparing the Edited Image against the Source Image. Ensure that the shape, texture, and structural details remain consistent, except for the parts explicitly targeted by the instruction. Label the result as Over Modification if the object is distorted, structurally damaged, or unnecessarily altered. Note that for object removal instruction, the deletion of non-target objects does not count as Over Modification. For object replacement instructions, if the target is successfully replaced with the requested object type and the replaced object itself is clearly recognizable as that object, alterations to surrounding objects are not classified as Over Modification. However, if the replaced object is not clearly recognizable, it should be classified as Over Modification. Furthermore, if blurriness makes it impossible to judge whether such damage occurred, treat it as Over Modification. If the edit is accurate and high-quality without these issues, label it as Flawless Execution.
**Note:** The target object may appear multiple times in an image. In such cases, each target instance must be evaluated independently following the above criteria, and the final score for the image should be the worst result among all target objects.

*Table 24.* The prompt used by Oracle-guide Mode to evaluate Instruction Following. (Part Three)

---

**Example**
Case 1:
Editing Instruction: Changed the color of the star painted on the tank from red to blue.
<Start Thinking>
First, I compare the Source Image and the Edited Image. Since these are cropped views of the star, I clearly see a difference: the star in the Source Image is red, while the star in the Edited Image is blue. This confirms a modification occurred.
Next, I check the Editing Instruction ("change... to blue"). The Edited Image shows the star is now blue, which aligns with the text instruction.
Finally, I verify Over Modification by comparing the Edited Image with the Reference Image and Source Image. The Reference Image shows a clean blue star that maintains the original painted texture and sharp geometric edges of the Source Image. However, in the Edited Image, the star's edges are jagged and blurry, and the texture looks like a plastic sticker rather than paint on metal. Although the color change is correct, the object's structural identity and texture are distorted compared to the Reference.
</Start Thinking>
<Start Final Answer>
Over Modification
</Start Final Answer>

Case 2:
Editing Instruction: Remove the orange cat sitting on the sofa.
<Start Thinking>
First, I confirm the modification: the orange cat is completely absent in the Edited Image, which aligns with the instruction.
Next, I verify Over Modification. I notice a non-target pillow was accidentally removed, and the sofa's shape is distorted. However, for removal instruction, the evaluation focuses strictly on the target. The accidental deletion or alteration of non-target objects does not constitute Over Modification. Since the target itself was successfully removed, these errors in the non-target areas are disregarded.
</Start Thinking>
<Start Final Answer>
Flawless Execution
</Start Final Answer>

Case 3:
Editing Instruction: Changed the color of the wheel on the yellow trailer from gray to red.
<Start Thinking>
First, I compare the Source Image and the Edited Image. I confirm that the wheels on the trailer have indeed turned red, so the primary modification action has been executed.
Next, I check the Editing Instruction ("...gray to red"). The instruction specifies changing the "gray" part, which semantically refers to the outer tires of the wheels, not the entire wheel structure.
Finally, I verify Over Modification by comparing the Edited Image with the Reference Image and Source Image. The Source Image shows the wheels have gray tires and orange hubs (rims). The Reference Image correctly depicts the intended result: only the gray tires are changed to red, while the orange hubs remain unchanged. However, in the Edited Image, the entire wheel structure—including the orange hubs—has been painted red. The model failed to disentangle the tire from the hub, resulting in an unwanted color change to the inner part of the target object. Since the hubs were not gray and should have been preserved, this constitutes an excessive alteration of the target object.
</Start Thinking>
<Start Final Answer>
Over Modification
</Start Final Answer>

**Your task is provided as follows**
Editing Instruction: {EDITING_INSTRUCTION}

---

*Table 25.* The prompt used by Oracle-guide Mode to evaluate Visual Consistency. (Part One)

You are a professional digital artist and image evaluation specialist. You will have to evaluate the effectiveness of the AI-generated image based on the given rules.

**Input Data Context:**
You will be provided with three images and one text instruction.
**CRITICAL NOTE:** To help you focus strictly on the background consistency, the specific target object mentioned in the instruction has been MASKED OUT (obscured with a white) in all provided images. Your Focus: Ignore the masked area completely. Look ONLY at the visible surrounding context (background, other objects). The Goal: Compare the Edited Image against the Source and Reference Images to detect any unintended changes.
1. **Source Image (Masked)**: The original image before modification (provided on the first).
2. **Edited Image (Masked)**: The edited version of the Source Image generated by the assistant (provided on the second).
3. **Reference Image (Masked)**: The "Gold Standard" showing exactly how the background should look. Use this as the ground truth for comparison (provide on the third).
4. **Editing Instruction**: The text command describing the modification.

**Evaluation Dimension: Visual Consistency**
Your primary goal is to assess whether the model preserved the integrity of the scene outside of the requested modification area. You must ignore the target object itself (which is evaluated separately) and focus strictly on the background environment and non-target objects. This involves two aspects:
1. **Global Scene Stability**: Did the model preserve the high-level semantic category of the environment and the artistic style? Focus exclusively on the general setting type (e.g., does a city remain a city? Does a bedroom remain a bedroom?). You must strictly IGNORE any changes to specific objects, layout, or visual details within the scene.
2. **Local Anomaly Detection**: Did the model strictly preserve the specific details of non-target objects without introducing anomalies?

**Evaluation Scale:**
**Perfect Consistency**: The highest standard. The background environment and all non-target objects remain visually identical to the Source Image. The edit integrates seamlessly without disturbing the surrounding pixels. The scene looks like the exact same photo, just with the specific target modified.
**Single Anomaly**: The general background environment remains consistent, but exactly ONE specific non-target object or detail has been altered, removed, or distorted. For example, everything is perfect except for one cup on the table that changed color, or the addition of one person who did not exist in the original image. Changes that only affect the overall image texture or global visual effects (e.g., lighting filters or grain) are not considered anomalies.
**Multiple Anomalies**: The general background environment remains consistent, but TWO OR MORE distinct non-target objects or details have been altered, removed, or distorted. For example, a painting on the wall changed content AND a chair in the corner disappeared. There are multiple scattered errors in the scene. Similarly, changes that only affect the overall image texture or global visual effects are not counted as anomalies.
**Scene Collapse**: The high-level semantic category of the environment or the artistic style has fundamentally changed. The Edited Image depicts a completely different type of location (e.g., a city turning into a forest) or implies a complete shift in medium (e.g., from photorealistic to an oil painting). Crucial: You must IGNORE changes to specific background objects or the layout of the scene; this label applies strictly when the general setting category or artistic style is destroyed.

**Evaluation Process:**
You should conduct the following assessment process: Use a chain-of-thought approach to think step by step, enclosing all detailed analysis within <Start Thinking>...</Start Thinking>tags. Then, you have two choices:
1. You can choose to use tools to assist your evaluation analysis. If you choose to call tools, output the tool call and then stop generating to wait for the tool's output.
2. You can choose to provide a final answer using the <Start Final Answer>...</Start Final Answer>tags. Strictly select one of the following descriptive labels according to the Scoring Criteria: Perfect Consistency, Single Anomaly, Multiple Anomalies, or Scene Collapse. Format your output as <Start Final Answer>Label</Start Final Answer>.

**Your can follow the evaluation steps below:**
1. **Check Global Scene Stability**: Begin by comparing the visible background of the Edited Image against the Source Image. Focus exclusively on the high-level semantic category of the environment and the artistic style. Ask yourself: Has the scene shifted to a completely different type of location (e.g., from a city to a forest) or a different artistic medium? You must strictly IGNORE changes to specific objects, layout, or lighting at this stage. If the general setting type or style has fundamentally changed, stop and immediately label it as Scene Collapse.

*Table 26.* The prompt used by Oracle-guide Mode to evaluate Visual Consistency. (Part Two)

2. **Scan and Count Local Anomalies**: If the global environment is preserved, meticulously scan every visible pixel outside the masked region. Since the target object is obscured by the mask, any visual discrepancy in the remaining area is considered a strict error. Compare the non-masked areas of the Edited Image with the Source Image to identify specific objects or details that are missing, distorted, additional, or changed. You must explicitly identify and count the number of distinct errors found (e.g., "Error 1: The lamp is missing. Error 2: An extra carpet was added unintentionally.").

3. **Determine Final Verdict**: Based on your explicit error count, select the final label. If the visible area is pristine with 0 errors, conclude Perfect Consistency. If you found exactly 1 distinct error, conclude Single Anomaly. If you found 2 or more distinct errors, conclude Multiple Anomalies.

**Example**
Case 1:
Editing Instruction: Remove the dog from the grass.
<Start Thinking>
The target (the dog) is masked out, so I am looking strictly at the surrounding grass field and the background fence. First, I Check Global Scene Stability. I compare the general setting of the Edited Image with the Source Image. The scene remains an outdoor grassy field environment, and the artistic style stays photorealistic. The location type has not shifted to a different setting (e.g., it didn't become a city or a room). No Scene Collapse.
Next, I Scan and Count Local Anomalies. I am scanning the visible areas for changes.
- Error 1: Looking at the fence in the top-left corner, I see that the Source Image has vertical slats, but in the Edited Image, the slats have turned diagonal.
- Error 2: Looking at the grass texture on the right side of the mask, there was a distinctive yellow flower in the Source Image that is completely gone in the Edited Image.
Count: I have identified 2 distinct errors in the non-target area.
Finally, I Determine Final Verdict. Since I found 2 distinct errors, this falls under the category of Multiple Anomalies.
</Start Thinking>
<Start Final Answer>
Multiple Anomalies
</Start Final Answer>

Case 2:
Editing Instruction: Change the car color to red.
<Start Thinking>
The user wants to change the car color. My task for "Visual Consistency" is to ensure that only the car changed and the rest of the street scene (buildings, road, trees) remains untouched. Global Check: The general environment type and artistic style are preserved. Local Check: Visually, the background looks stable, but I need to be precise about "Single" vs "Multiple" Anomalies. There might be distortions in the complex building windows or the pavement. To accurately count the number of errors in non-target areas, I need to detect the specific regions where these distortions occurred.
</Start Thinking>
<tool_call>
{ "name": "localize_differences", "parameters": { "comparison_image_1": "Source Image", "comparison_image_2": "Edited Image" } }
</tool_call>

**Note**
1. Each evaluation step must start with <Start Thinking>...</Start Thinking>tags to write detailed assessments, followed by an <tool_call>...</tool_call>tags (If you decide to use tools, provide the function name and parameters in JSON format. You can call multiple tools at once.) or <Start Final Answer>...</Start Final Answer>tags (If you decide to output evaluation result), but never both.
2. If you choose to use tools, please wait for the tool results to be returned in the next round. Do not attempt to generate or predict tool outputs yourself. Only call a tool when you are unable to make a reliable judgment based on the available information; if you can confidently make a decision without assistance, do not use tools for verification.
3. If a single reasoning step requires tool calls on multiple images, please output all necessary tool calls together in one <tool_call>...</tool_call>tags. Do not split these into separate steps; instead, issue simultaneous tool calls for each image as needed.
4. The localize_differences tool detects global changes. Please note that these detections are based on strict pixel-level comparison and might include negligible variations imperceptible to humans. You should disregard insignificant fluctuations and only focus on the crops showing significant, visually obvious changes.

**Your task is provided as follows**
Editing Instruction: {EDITING_INSTRUCTION}

*Table 27.* The prompt used by LMM-as-a-Judge to evaluate Instruction Following. (Part One)

You are a professional digital artist and image evaluation specialist. You will have to evaluate the effectiveness of the AI-generated image based on the given rules.

**Input Data Context:**
You will be provided with three images and one text instruction.
1. **Source Image**: The original image before modification (provided on the first)
2. **Edited Image**: The edited version of the Source Image generated by the assistant (provided on the second.
3. **Editing Instruction**: The text command describing the modification.

**Evaluation Dimension: Instruction Following**
Your primary goal is to assess whether the Edited Image faithfully followed the user's edit instruction to modify the Source Image. This involves three aspects:
1. **Localization**: Did the model perform the modification at the correct target location?
2. **Action Execution**: Did the model perform the correct modification action (e.g., replace, remove, alter) on that object as requested? (e.g., If the instruction is "change color", did the color actually change to the target color?)
3. **Over Modification**: Did the model preserve the original identity and details of the target object that were not specified to change?
**Note:** Do not assess any unintended modifications beyond the instruction. Focus strictly on the target region and the requested action. Issues such as visual consistency or overall image quality should NOT negatively impact the score. Such evaluations fall under separate criteria (e.g., visual consistency and visual quality).

**Evaluation Scale:**
**Flawless Execution**: The model demonstrates flawless instruction following. It correctly identifies the target region or object, executes the specific modification accurately, and strictly preserves the object's original identity (including shape, texture, and structure) without introducing any unintended changes.
**Over Modification**: This label applies when the model correctly locates the target and executes the requested edit, but fails to preserve the object's original details that should have remained unchanged. Except for the specific changes explicitly requested by the instruction, the target object must remain visually identical to the Source Image; therefore, any unrequested alterations to the object's structure, or fine details—such as changing a T-shirt into a hoodie when only a color shift was requested—are classified as Over Modification. Notably, for object removal instructions, the accidental deletion of non-target objects is NOT penalized under this category. For object replacement instructions, as long as the target is successfully replaced with the requested type of object and the replaced object itself is clearly recognizable as that object, alterations to surrounding objects are NOT classified as Over Modification. However, if the replaced object is not clearly recognizable, it should be classified as Over Modification. Additionally, if the new object is inconsistent with the original image's style (e.g., lighting, rendering style, or realism level), it should also be classified as Over Modification.

*Table 28.* The prompt used by LMM-as-a-Judge to evaluate Instruction Following. (Part Two)

**Wrong Action**: The model successfully locates the target object but executes a modification that mismatches the requested action category. You must strictly verify against these specific categories: Change Color, Change Material, Change Text, Change Shape, Object Count (Reduction, Addition), and Object Manipulation (Remove, Replace). If the model performs an action from a different category than requested (e.g., executing a "Remove Object" operation when a "Change Color" was requested, or "Replace Object" when only a "Change Material" was asked), it must be labeled as Wrong Action. Notably, for object count reduction instructions, if the target objects are not reduced to the exact specified quantity, this is classified as Wrong Action; similarly, for object addition instructions, if the target objects are not increased to the exact specified quantity, this is also classified as Wrong Action. For the object removal instructions, filling in the area where the object was located after removal shall not be considered a Wrong Action, unless it is clearly evident that a replacement operation has been performed.

**Localization Failure**: Localization Failure occurs when the model fails to execute the specified modification on the intended target, leaving the object effectively unchanged. This score encompasses scenarios where the target exhibits minor, unintentional artifacts—such as subtle color shifts or slight geometric distortions—that do not constitute the requested edit, as well as cases where severe blurriness or distortion renders visual verification of the modification impossible. Furthermore, this score also includes mis-localization within the target object, where the model modifies an incorrect sub-component of the target object (e.g., when the target object is a wheel and the instruction is to change the hub of the wheel to red, but the model instead changes the tire to red), because the specific attribute defined by the instruction remains unmodified.

**Evaluation Process:**
Use a chain-of-thought approach, enclosing your detailed analysis within <Start Thinking>...</Start Thinking>tags. Use <Start Final Answer>...</Start Final Answer>tags to provide a final answer. Strictly select one of the following descriptive labels according to the Scoring Criteria: Flawless Execution, Over Modification, Wrong Action, or Localization Failure. You must strictly follow this logical sequence:
1. **Verify Modification Occurrence**: First, compare the Source Image and the Edited Image. Do not rely on mere visual differences or pixel-level changes. Instead, focus on whether the intended edit has effectively occurred. Even if the image exhibits changes (such as artifacts, color shifts, or slight distortions), if the target object remains identifiable or has not undergone the requested modification, you must label it as Localization Failure. Additionally, if the target is too blurry to definitively verify the modification, label it as Localization Failure. If the model performs any modification on an incorrect sub-component within the target object, and the specific part or attribute specified in the instruction is not correctly modified, label it as Localization Failure.
2. **Verify Action Alignment**: If a change occurred, check if it matches the Editing Instruction. Determine if the specific action (e.g., color change, object removal) was performed correctly. If the model performed a modification that contradicts the instruction (e.g., turning an object green instead of red), label it as Wrong Action. Notably, for object count reduction instructions, if the target objects are not reduced to the exact specified quantity, this is classified as Wrong Action; similarly, for object addition instructions, if the target objects are not increased to the exact specified quantity, this is also classified as Wrong Action.
3. **Verify Over Modification**: Finally, assess if the object's identity was preserved by comparing the Edited Image against the Source Image. Ensure that the shape, texture, and structural details remain consistent, except for the parts explicitly targeted by the instruction. Label the result as Over Modification if the object is distorted, structurally damaged, or unnecessarily altered. Note that for object removal instruction, the deletion of non-target objects does not count as Over Modification. For object replacement instructions, if the target is successfully replaced with the requested object type and the replaced object itself is clearly recognizable as that object, alterations to surrounding objects are not classified as Over Modification. However, if the replaced object is not clearly recognizable, it should be classified as Over Modification. Furthermore, if blurriness makes it impossible to judge whether such damage occurred, treat it as Over Modification. If the edit is accurate and high-quality without these issues, label it as Flawless Execution.
**Note:** The target object may appear multiple times in an image. In such cases, each target instance must be evaluated independently following the above criteria, and the final score for the image should be the worst result among all target objects.

*Table 29.* The prompt used by LMM-as-a-Judge to evaluate Instruction Following. (Part Three)

---

**Example**
Case 1:
Editing Instruction: Changed the color of the star painted on the tank from red to blue.
<Start Thinking>
First, I compare the Source Image and the Edited Image. I clearly see a difference: the star in the Source Image is red, while the star in the Edited Image is blue. This confirms a modification occurred.
Next, I check the Editing Instruction ("change... to blue"). The Edited Image shows the star is now blue, which aligns with the text instruction.
Finally, I verify Over Modification by comparing the Edited Image with the source image. In the Edited Image, the star's edges are jagged and blurry, and the texture looks like a plastic sticker rather than paint on metal. Although the color change is correct, the object's structural identity and texture are distorted compared to the Source Image.
</Start Thinking>
<Start Final Answer>
Over Modification
</Start Final Answer>

Case 2:
Editing Instruction: Remove the orange cat sitting on the sofa.
<Start Thinking>
First, I confirm the modification: the orange cat is completely absent in the Edited Image, which aligns with the instruction.
Next, I verify Over Modification. I notice a non-target pillow was accidentally removed, and the sofa's shape is distorted. However, for removal instruction, the evaluation focuses strictly on the target. The accidental deletion or alteration of non-target objects does not constitute Over Modification. Since the target itself was successfully removed, these errors in the non-target areas are disregarded.
</Start Thinking>
<Start Final Answer>
Flawless Execution
</Start Final Answer>

Case 3:
Editing Instruction: Changed the color of the wheel on the yellow trailer from gray to red.
<Start Thinking>
First, I compare the Source Image and the Edited Image. I confirm that the wheels on the trailer have indeed turned red, so the primary modification action has been executed.
Next, I check the Editing Instruction ("...gray to red"). The instruction specifies changing the "gray" part, which semantically refers to the outer tires of the wheels, not the entire wheel structure.
Finally, I verify Over Modification by comparing the Edited Image with the Source Image. The Source Image shows the wheels have gray tires and orange hubs (rims). In the Edited Image, the entire wheel structure—including the orange hubs—has been painted red. The model failed to disentangle the tire from the hub, resulting in an unwanted color change to the inner part of the target object. Since the hubs were not gray and should have been preserved, this constitutes an excessive alteration of the target object.
</Start Thinking>
<Start Final Answer>
Over Modification
</Start Final Answer>

**Your task is provided as follows**
Editing Instruction: {EDITING_INSTRUCTION}

---

*Table 30.* The prompt used by LMM-as-a-Judge to evaluate Visual Consistency. (Part One)

You are a professional digital artist and image evaluation specialist. You will have to evaluate the effectiveness of the AI-generated image based on the given rules.

**Input Data Context:**
You will be provided with three images and one text instruction.
1. **Source Image**: The original image before modification (provided on the first).
2. **Edited Image**: The edited version of the Source Image generated by the assistant (provided on the second)
3. **Editing Instruction**: The text command describing the modification.

**Evaluation Dimension: Visual Consistency**
Your primary goal is to assess whether the model preserved the integrity of the scene outside of the requested modification area. You must ignore the target object itself (which is evaluated separately) and focus strictly on the background environment and non-target objects. This involves two aspects:
1. **Global Scene Stability**: Did the model preserve the high-level semantic category of the environment and the artistic style? Focus exclusively on the general setting type (e.g., does a city remain a city? Does a bedroom remain a bedroom?). You must strictly IGNORE any changes to specific objects, layout, or visual details within the scene.
2. **Local Anomaly Detection**: Did the model strictly preserve the specific details of non-target objects without introducing anomalies?

**Evaluation Scale:**
**Perfect Consistency**: The highest standard. The background environment and all non-target objects remain visually identical to the Source Image. The edit integrates seamlessly without disturbing the surrounding pixels. The scene looks like the exact same photo, just with the specific target modified.
**Single Anomaly**: The general background environment remains consistent, but exactly ONE specific non-target object or detail has been altered, removed, or distorted. For example, everything is perfect except for one cup on the table that changed color, or the addition of one person who did not exist in the original image. Changes that only affect the overall image texture or global visual effects (e.g., lighting filters or grain) are not considered anomalies.
**Multiple Anomalies**: The general background environment remains consistent, but TWO OR MORE distinct non-target objects or details have been altered, removed, or distorted. For example, a painting on the wall changed content AND a chair in the corner disappeared. There are multiple scattered errors in the scene. Similarly, changes that only affect the overall image texture or global visual effects are not counted as anomalies.
**Scene Collapse**: The high-level semantic category of the environment or the artistic style has fundamentally changed. The Edited Image depicts a completely different type of location (e.g., a city turning into a forest) or implies a complete shift in medium (e.g., from photorealistic to an oil painting). Crucial: You must IGNORE changes to specific background objects or the layout of the scene; this label applies strictly when the general setting category or artistic style is destroyed.

**Your can follow the evaluation steps below:**
1. **Check Global Scene Stability**: Begin by comparing the visible background of the Edited Image against the Source Image. Focus exclusively on the high-level semantic category of the environment and the artistic style. Ask yourself: Has the scene shifted to a completely different type of location (e.g., from a city to a forest) or a different artistic medium? You must strictly IGNORE changes to specific objects, layout, or lighting at this stage. If the general setting type or style has fundamentally changed, stop and immediately label it as Scene Collapse.

*Table 31.* The prompt used by LMM-as-a-Judge to evaluate Visual Consistency. (Part Two)

2. **Scan and Count Local Anomalies**: If the global environment is preserved, meticulously scan every visible pixel outside the masked region. Since the target object is obscured by the mask, any visual discrepancy in the remaining area is considered a strict error. Compare the non-masked areas of the Edited Image with the Source Image to identify specific objects or details that are missing, distorted, additional, or changed. You must explicitly identify and count the number of distinct errors found (e.g., "Error 1: The lamp is missing. Error 2: An extra carpet was added unintentionally.").

3. **Determine Final Verdict**: Based on your explicit error count, select the final label. If the visible area is pristine with 0 errors, conclude Perfect Consistency. If you found exactly 1 distinct error, conclude Single Anomaly. If you found 2 or more distinct errors, conclude Multiple Anomalies.

**Example**
Case 1:
Editing Instruction: Remove the dog from the grass.
<Start Thinking>
The target (the dog) is removed in the Edited Image, so I am looking strictly at the surrounding grass field and the background fence.
First, I Check Global Scene Stability. I compare the general setting of the Edited Image with the Original Image. The scene remains an outdoor grassy field environment, and the artistic style stays photorealistic. The location type has not shifted to a different setting (e.g., it didn't become a city or a room). No Scene Collapse.
Next, I Scan and Count Local Anomalies. I am scanning the visible areas for changes.
- Error 1: Looking at the fence in the top-left corner, I see that the Original Image has vertical slats, but in the Edited Image, the slats have turned diagonal.
- Error 2: Looking at the grass texture on the right side of where the dog used to be, there was a distinctive yellow flower in the Original Image that is completely gone in the Edited Image.
Count: I have identified 2 distinct errors in the non-target area. Finally, I Determine Final Verdict. Since I found 2 distinct errors, this falls under the category of Multiple Anomalies.
</Start Thinking>
<Start Final Answer>
Multiple Anomalies
</Start Final Answer>

**Your task is provided as follows**
Editing Instruction: {EDITING_INSTRUCTION}

