# OpenReview forum: "DLEBench: Evaluating Small-scale Object Editing Ability for Instruction-based Image Editing Model"
_ICML.cc/2026/Conference — ICML 2026 regular_

### Official Review · Reviewer_GhMP · 2026-03-11

**Soundness:** 3
**Presentation:** 3
**Significance:** 2
**Originality:** 2
**Overall Recommendation:** 4
**Confidence:** 4

**Summary:**

The paper introduces DLEBench, a benchmark specifically designed to evaluate the small-scale object editing ability of instruction-based image editing models. The benchmark contains 1,889 samples spanning 7 instruction categories, with a particular focus on challenging cases where target objects occupy only a small portion of the image and may involve occlusion or multi-object scenarios. The paper also proposes refined scoring rubrics for Instruction Following (IF) and Visual Consistency (VC), together with two evaluation modes, Tool-driven and Oracle-guided, and conducts a systematic evaluation on 10 representative models. The experimental results show substantial performance gaps in small-object editing, highlighting the need for specialized benchmarks to better assess and improve this capability.

**Compliance With Llm Reviewing Policy:**

Affirmed.

**Final Justification:**

My concerns have been sufficiently addressed, and I will raise my score to weak accept.

**Key Questions For Authors:**

N/A

**Limitations:**

yes

**Strengths And Weaknesses:**

Strengths
1. The benchmark construction pipeline is relatively systematic. The paper presents a clear three-stage transformation process from visual reasoning data to image editing samples, with an additional human verification stage to improve data quality.
2. The experimental coverage is reasonably broad. The authors evaluate 10 models spanning closed-source, open-source, and different architectural families, and provide analyses across instruction types, target scales, and evaluation dimensions, which makes the empirical study reasonably informative.

Weaknesses
1. The motivation regarding the insufficiency of existing benchmarks is not fully substantiated. The paper mainly argues for the difference through comparisons of target-area distributions, but does not provide more direct evidence that models that perform strongly on prior benchmarks systematically fail on small-object editing, nor that subsets of existing benchmarks cannot already capture such failures.
2. The experimental section lacks stronger ablations. For example, it is unclear how much each component in the three-stage construction pipeline contributes, including GPT-4.1-based metadata generation, crop-and-edit reference generation, and manual correction; similarly, the sensitivity of the adaptive bbox expansion parameters is not studied. Without such ablations, it is difficult to assess whether these design choices are truly necessary.
3. The fairness and generalizability of the Tool-driven mode are not sufficiently clear. This mode depends on a specific toolchain (e.g., GroundingDINO, difference localization, super-resolution), and its evaluation quality may be tightly coupled to those tool implementations. The paper does not sufficiently establish whether the conclusions remain stable when using different tools or a different backbone LMM.
4. The data distribution is clearly imbalanced. Categories such as color, OCR, and removal dominate the benchmark, while shape, count, and replacement are relatively underrepresented.

---

> ### Author Rebuttal · Authors · 2026-03-31
>
> Thank you for your time and feedback. We conducted additional analyses and experiments to address your concerns.
>
> >W1: Insufficiency of existing benchmarks is not fully substantiated.
>
> A: We thank the reviewer for this critique. Beyond target-area distributions, we provide two additional pieces of evidence.
>
> **1. Existing benchmark subsets are too small**
>
> We count samples in different target-area ranges across five datasets:
>
> ### Table 1: Absolute sample counts
> |Benchmark|0%-1%|1%-10%|>10%|
> |-|-|-|-|
> |DLEBench|1575|314|0|
> |RISE|22|93|245|
> |KRIS|8|220|1039|
> |UniREditBench|4|223|2056|
> |ImageEdit|3|34|478|
>
> Prior benchmarks contain only 3-22 samples in the extreme small-object range (0%-1%), which is too few to form a reliable subset for evaluating such failures.
>
> **2. Strong performance on prior benchmarks does not transfer to DLEBench**
>
> We evaluate Gemini-3-Pro on two recent benchmarks, KRIS and UniREditBench. Gemini-3-Pro achieves overall scores of 86.61 and 80.35, respectively, higher than all results reported in the original papers. However, its performance drops to 65.55 on DLEBench, with IF further decreasing to 48.97 (Table 2). This shows that strong benchmark performance does not necessarily transfer to small-object editing.
>
> ### Table 2: Gemini-3-Pro across existing benchmarks and DLEBench.
> |Benchmark|VC|IF|Overall|
> |-|-|-|-|
> |KRIS|88.30|84.91|86.61|
> |UniREditBench|-|-|80.35|
> |DLEBench|82.13|48.97|65.55|
>
> >W2: Lack of ablation studies to justify the dataset construction and parameter choices.
>
> A: Thank you for raising this suggestion. We added ablations on both the data pipeline and adaptive bbox expansion.
>
> **1. GPT-4.1 metadata generation**
>
> We compared the initial GPT-4.1-generated instructions with the final human-verified versions and found that 89.3% were retained without manual revision. This indicates that GPT-4.1 reliably performs metadata generation.
>
> **2. Crop-and-edit reference generation and adaptive bbox expansion**
>
> To evaluate the necessity of crop-and-edit and adaptive bbox expansion, we compare our method against three baselines: (i) Full Editing, which edits the full image directly without cropping; (ii) fixed expansion with $\lambda=0.3$; and (iii) fixed expansion with $\lambda=6.0$. We evaluate these variants on 100 uniformly sampled images from DLEBench under the reference-free Oracle setting.
>
> ### Table 3: Ablation on crop-and-edit and bbox expansion
> |Method|IF|VC|
> |-|-|-|
> |Full Editing|45.00|94.04|
> |$\lambda$= 0.3|81.00|83.16|
> |$\lambda$= 6.0|78.00|92.00|
> |Ours|89.33|90.00|
>
> These results show that both choices are necessary: without crop-and-edit, small objects are often not properly localized (IF 45.00), while fixed expansion ratios lead to either insufficient context ($\lambda=0.3$, lower VC) or excessive context ($\lambda=6.0$, lower IF). Our adaptive strategy provides the best trade-off between IF and VC.
>
> >W3: The fairness and generalizability of the Tool-driven mode.
>
> A: Thank you for raising this comment. To assess fairness and generalizability, we replace both the evaluator backbone and learned tools in Tool-driven mode. We use GPT-4.1 instead of Gemini, replace GroundingDINO(GD) with YOLO-World(YW)[1], and replace Real-ESRGAN(ESR) with SwinIR(SIR)[2]. Difference and Zoom-in are kept unchanged since they are deterministic image operations.
>
> ### Table 4: Tool-driven mode under tool/backbone substitutions. S=Spearman, P=Pearson.
> |Method|IF-S↑|IF-P↑|IF-MAE↓|VC-S↑|VC-P↑|VC-MAE↓|
> |-|-|-|:-:|-|-|:-:|
> |w/o Tool|0.39|0.44|1.14|0.39|0.37|0.97|
> |GD+ESR|0.54|0.56|0.77|0.52|0.50|0.69|
> |GD+SIR|0.54|0.55|0.77|0.53|0.51|0.68|
> |YW+ESR|0.51|0.53|0.80|0.52|0.51|0.69|
> |YW+SIR|0.50|0.52|0.81|0.51|0.49|0.70|
>
> Table 4 shows that all tool-based variants consistently outperform the tool-less baseline, despite these substitutions. This indicates that the improvement is not tied to a specific toolchain, but comes from the visual feedback itself.
>
> >W4: Imbalanced instruction distribution.
>
> A: Thank you for raising this concern. As noted in the Limitations section, we acknowledge that the instruction distribution in DLEBench is imbalanced.
>
> We construct DLEBench from visually dense VQA datasets for two reasons. First, **they naturally provide precise spatial descriptions for localizing small-scale objects, which are essential for small-object editing**; otherwise, the target can become ambiguous and difficult to locate. Second, **their questions about object categories and attributes can be naturally transformed into editing instructions.** We therefore reuse these high-quality resources instead of relying entirely on manual annotation, although manual annotation could improve category balance.
>
> This choice inevitably inherits the source datasets' distribution bias. As noted in Future Work, we will explore data augmentation to improve category balance in future versions.
>
> [1] YOLO-World: Real-Time Open-Vocabulary Object Detection.
>
> [2] SwinIR: Image Restoration Using Swin Transformer.

---

> > ### Author Rebuttal · Reviewer_GhMP · 2026-04-03
> >
> > Thank you to the authors for the detailed rebuttal. My concerns have been sufficiently addressed, and I will raise my score to weak accept.

---

> > > ### Author Response · Authors · 2026-04-05
> > >
> > > Dear Reviewer GhMP,
> > >
> > > We sincerely thank you for your positive feedback and for raising the score. We are glad that our clarifications and additional experiments have sufficiently addressed your concerns. We greatly appreciate your time and careful review.
> > >
> > > Best regards,
> > >
> > > The Authors

---

### Official Review · Reviewer_7B8b · 2026-03-11

**Soundness:** 3
**Presentation:** 2
**Significance:** 3
**Originality:** 3
**Overall Recommendation:** 4
**Confidence:** 3

**Summary:**

This paper presents DLEBench, a benchmark for evaluating small-object editing in instruction-based image editing models. It includes 1,889 samples across 7 instruction types, together with a hierarchical IF/VC scoring scheme and two evaluation modes: Tool-driven and Oracle-guided. Experiments show that current models still struggle with small-object localization, precise editing, and avoiding unnecessary modifications.

**Compliance With Llm Reviewing Policy:**

Affirmed.

**Final Justification:**

The paper presents a useful benchmark for small-scale object editing with a reasonably solid evaluation protocol. The authors’ rebuttal address my main concerns, so I keep my final recommendation at **Weak Accept**.

**Key Questions For Authors:**

**1. Source-data bias.** Since DLEBench is converted from VQA samples rather than real user editing requests, how do the authors assess its distribution bias and external validity?

**2. Reference generation.** Since the reference images are produced by Gemini-3-Pro with crop-and-edit and then filtered by humans, how do the authors rule out bias toward this specific reference-generation pipeline?

**3. Evaluator bias.** How do the authors rule out evaluator-model coupling in the automatic assessment pipeline, especially when similar model preferences or representations may exist?

Additional evidence and analysis would  affect my view of the benchmark’s generality and reliability.

**Limitations:**

Yes.

**Strengths And Weaknesses:**

**Strengths**

**1. Clear problem focus.** This paper targets the long-overlooked capability of small-object editing in instruction-based image editing. Compared with existing benchmarks that mainly emphasize salient foreground objects, DLEBench specifically evaluates small-object localization, fine-grained local editing, and the ability to avoid undesired changes outside the target region, which helps reveal important failure modes of current models more clearly.

**2. Systematic benchmark design.** The paper goes beyond simply a dataset: it introduces a pipeline for converting visual reasoning samples into editing tasks, a crop-and-edit strategy for generating reference images, and a structured evaluation protocol with hierarchical IF/VC criteria and two modes, Tool-driven and Oracle-guided.

**Weaknesses**

**1. Source-data bias.** DLEBench is not derived from real user editing requests, but from counterfactual transformations of VQA samples from visual reasoning benchmarks. As a result, its task distribution is heavily shaped by the upstream data sources and does not fully reflect open-world editing demands. Although the paper discusses instruction imbalance, it does not sufficiently address this more fundamental source-data bias, which limits its claim as a general benchmark for small-object editing.

**2. Reference-generation bias.** The reference images are generated using Gemini-3-Pro + crop-and-edit, followed by human filtering. This design introduces clear risks: crop-and-edit changes the original context of the task and may impose specific preferences in local blending, boundary handling, or visual style. It remains unclear whether model performance truly reflects small-object editing ability, or is partly influenced by similarity to the reference-generation paradigm.

**3. Evaluator coupling.** The paper uses a strong multimodal model as the evaluation backbone and supports its validity with human agreement experiments. However, if evaluated models share similar preferences or visual-language representations with the evaluator, this may still introduce bias. In fine-grained editing tasks in particular, the evaluator could implicitly favor certain generation styles or editing paths.

---

> ### Author Rebuttal · Authors · 2026-03-31
>
> Thanks for your review, feedback, and positive assessment of our work.
>
> >W1 & Q1: Concerns regarding source-data bias.
>
> A: Thank you for raising this concern. To assess the distribution bias and external validity of DLEBench (DLE), we compared it with two widely used datasets of user-generated instructions: HumanEdit (HE)[1] and MagicBrush (MB)[2], from two perspectives: semantic similarity and instruction length.
>
> **1. Semantic Similarity**
>
> To examine whether DLEBench shares similar editing intent with user-generated datasets, we encode all instructions using Sentence-BERT (all-mpnet-base-v2) and compute centroid similarity and average sample similarity.
>
> ### Table 1: Semantic Similarity
> |Methods|Centroid Similarity|Average Sample Similarity|
> |-|:-:|:-:|
> |DLE vs HE|0.8033|0.2378|
> |DLE vs MB|0.7609|0.2093|
> |HE vs MB|0.8969|0.2025|
>
> The centroid similarity between DLEBench and the two user-generated datasets is comparable to that between HE and MB, suggesting that DLEBench captures a similar high-level editing intent distribution. The low sample-level similarity across all pairs reflects the natural diversity of user phrasing.
>
> **2. Instruction Length**
>
> We compare average instruction length and find that DLEBench (12.5 words) is longer than HE (5.6) and MB (6.5). We attribute this mainly to the fine-grained localization in small-object editing. In cluttered scenes, users often need attributes and spatial cues to disambiguate the target object. This is further validated by Table 2, which shows a higher frequency of spatial and attribute words per instruction in DLEBench.
>
> ### Table 2: Average Frequencies.
> |Dataset|Attribute Freq.|Spatial Freq.|
> |-|:-:|:-:|
> |DLE|0.675|0.139|
> |HE|0.235|0.025|
> |MB|0.178|0.04|
>
> Overall, these results suggest that DLEBench has meaningful overlap with real user editing requests, while its longer and more specific instructions reflect the localization demands of small-object editing.
>
> [1]HumanEdit: A High-Quality Human-Rewarded Dataset for Instruction-based Image Editing.
>
> [2]MagicBrush: A Manually Annotated Dataset for Instruction-Guided Image Editing.
>
> >W2 & Q2: Evaluation bias stemming from reference images.
>
> A: We thank the reviewer for raising this important question. We agree that references generated by specific methods could introduce bias. To address this, we clarify our protocol and provide additional evidence.
>
> **First, Oracle evaluation is not based on matching to the reference image.**
> Instead, Oracle scores edits according to the designated evaluation procedure grounded in the text instruction(Line 2112). The reference image is only auxiliary, rather than the primary basis for judgment.
>
> **Second, Oracle shows strong agreement with human evaluators who do not see reference images.**
> As reported in Table 3, Oracle (w/ reference) aligns closely with human judgments, although humans use the same criteria without access to references. This suggests that Oracle is not simply favoring artifacts or styles from the reference-generation pipeline.
>
> **Third, we performed an ablation that removes reference images entirely.**
> As shown in Table 3, even without references, Oracle still maintains high agreement with human judgments. This indicates that Oracle does not rely on the reference-generation pipeline and that reference-induced bias is limited.
>
> ### Table 3: Alignment with Human
> |Method|Criteria|Spearman↑|Pearson↑|MAE↓|
> |-|-|:-:|:-:|-|
> |w/ reference|IF|0.84|0.85|0.28|
> ||VC|0.83|0.86|0.21|
> |w/o reference|IF|0.82|0.84|0.30|
> ||VC|0.84|0.86|0.21|
>
> >W3 & Q3: Concerns regarding evaluator-model coupling.
>
> A: Thank you for raising this suggestion. We agree that potential evaluator-model coupling is an important concern for automatic evaluation. To mitigate this issue, our framework adopts a **predefined evaluation procedure grounded in the text instruction (Line 2112), together with mode-specific constraints**, rather than relying on unconstrained judgments from the evaluator. In particular, Tool mode incorporates external tool feedback, while Oracle mode provides a pre-cropped target object as direct evidence for assessment. These design choices reduce the evaluator’s degrees of freedom and make the decision process more evidence-driven.
>
> To examine evaluator dependence, we replaced Gemini-3-Pro with GPT-4.1 as the evaluator in both modes and compared the results with a standard prompt-engineering-only baseline.
>
> ### Table 4: Alignment using GPT-4.1 as the evaluator. S=Spearman, P=Pearson.
> |Method|IF-S↑|IF-P↑|IF-MAE↓|VC-S↑|VC-P↑|VC-MAE↓|
> |-|-|-|:-:|-|-|:-:|
> |Baseline|0.39|0.44|1.14|0.39|0.37|0.97|
> |Tool|0.54|0.56|0.77|0.52|0.50|0.69|
> |Oracle (w/o reference)|0.67|0.69|0.54|0.65|0.64|0.50|
>
> Since both modes still outperform the baseline with a different evaluator, the gains are not mainly due to a specific evaluator model. While shared biases may still exist, this suggests that our framework is robust across evaluators and less sensitive to evaluator-specific preferences.

---

> > ### Author Rebuttal · Reviewer_7B8b · 2026-04-02
> >
> > Thank you for the detailed rebuttal. My main concerns have been adequately addressed. I believe this paper merits a weak accept, and therefore I keep my final score unchanged.

---

> > > ### Author Response · Authors · 2026-04-05
> > >
> > > Dear Reviewer 7B8b,
> > >
> > > Thank you very much for your positive evaluation and encouraging feedback. We are glad that our rebuttal has adequately addressed your main concerns. We sincerely appreciate your time, expertise, and careful review of our paper.
> > >
> > > Best regards,
> > >
> > > The Authors

---

### Official Review · Reviewer_kEx7 · 2026-03-11

**Soundness:** 3
**Presentation:** 3
**Significance:** 2
**Originality:** 2
**Overall Recommendation:** 4
**Confidence:** 3

**Summary:**

This paper introduces DLEBench, a benchmark comprising 1,889 samples and seven instruction types, to address the lack of systematic evaluation for small-scale object editing. The dataset is constructed via a three-stage semi-automated pipeline: (1) transforming VQA questions and options into editing instructions; (2) employing a crop-and-edit strategy with Gemini-3-Pro to generate localized reference images; (3) conducting human verification. The evaluation protocol defines two dimensions: Instruction Following (IF), which assesses the success of target editing, and Visual Consistency (VC), which evaluates the preservation of non-target regions, both utilizing hierarchical scoring.  Furthermore, the paper introduces a Tool-driven Mode that uses external tools for diagnostic scoring, and an Oracle-guided Mode that uses human-annotated bounding boxes to assist the model's judgment. Through extensive experiments on 10 representative models, the study reveals significant performance gaps in current models when tasked with fine-grained, small-scale image editing, highlighting the need for specialized benchmarks to advance this field.

**Compliance With Llm Reviewing Policy:**

Affirmed.

**Final Justification:**

My concerns have been fully addressed.

**Key Questions For Authors:**

See Weaknesses.

**Limitations:**

Yes.

**Strengths And Weaknesses:**

Strengths:
1. This work is the first to focus on the lack of small-scale object editing capabilities in IIEMs, filling the gap in the evaluation of fine-grained local editing in existing benchmarks, and has strong practical value.
2. DLEBench proposes IF and VC evaluation dimensions and designs hierarchical scoring criteria, which can clearly diagnose the specific bottlenecks of the model in small-scale image editing.
3. The experimental analysis is comprehensive, covering different instruction types, scales, failure modes, and consistency with human evaluation.

Weaknesses:
1. The instruction types in the test samples are overly simplistic, lacking many common editing operations such as moving, scaling, and rotating objects. Furthermore, the benchmark fails to account for multi-instruction coupling scenarios common in real-world applications (e.g., simultaneously changing both material and color), which limits a deeper exploration of task difficulty scaling.
2. The evaluation exhibits potential bias, as using Gemini-3-Pro to generate the "Golden Answers" while treating them as the gold standard in Oracle mode may grant the model a "home-field advantage," leading to inflated performance scores.
3. This benchmark lacks consideration for the diversity of answers. The image editing is highly open-ended, and using only a single reference image in Oracle mode comparison may misjudge excellent models.
4. Although this work identified the shortcomings of various models through experiments, it lacked rigorous theoretical analysis of the underlying reasons for the differences in performance between models, and it also lacked discussion on how to systematically improve small-scale editing capabilities in the future.

---

> ### Author Rebuttal · Authors · 2026-03-31
>
> Thank you very much for your valuable time and feedback. We provide the following clarifications to address your concerns.
>
> >W1: The instruction types are overly simplistic.
>
> A: We thank the reviewer for this comment. We agree that the current instruction set in DLEBench does not yet cover more complex editing operations.
>
> As noted in the Limitations section, this mainly stems from our data construction pipeline. DLEBench is built from VQA datasets, whose questions primarily focus on object categories and attributes, while expressions describing geometric edits or coupled instructions are rare. As a result, the current benchmark mainly covers category- and attribute-level local edits, with limited support for more complex scenarios.
>
> Despite this limitation, we argue this is a pragmatic trade-off. We build on visually dense VQA data because **small-scale object editing requires unambiguous localization of tiny targets, and such datasets naturally provide precise spatial grounding.** Repurposing them also enables scalable construction of a high-quality benchmark, avoiding the prohibitive cost of collecting and annotating complex editing instructions from scratch.
>
> Furthermore, our results show that even under the current instruction setting, state-of-the-art models still achieve limited performance, which highlights the value of DLEBench in evaluating this challenging capability.
>
> **We agree that richer editing operations are important, and we will extend the benchmark to include more complex editing instructions in future work.**
>
> >W2: Reference image causes potential bias.
>
> A: We thank the reviewer for raising this important concern. We clarify that, in our Oracle mode, the generated golden answer is **not used as an exact target** for direct comparison. Instead, the evaluation is conducted according to our predefined protocol based on the **text instruction and evaluation criteria** (Line 2112). Therefore, the Oracle does not simply reward outputs for resembling the Gemini-generated reference image.
>
> Moreover, the high agreement between Oracle (w/ reference) and human judgments in Table 1 supports this claim. Human annotators follow the same evaluation criteria but are **not provided with the reference images** (the annotation interface is shown in Line 883). Hence, their judgments are independent of the generated golden answers. The strong Oracle-human alignment suggests that the Oracle primarily captures instruction satisfaction rather than matching the reference image.
>
> To further address this concern, we conducted an ablation where the reference images were completely removed from Oracle mode (w/o reference).
>
> ### Table 1: Alignment with Human Judgments
> |Method|Criteria|Spearman↑|Pearson↑|MAE↓|
> |-|:-:|:-:|:-:|:-:|
> |w/ reference|IF|0.84|0.85|0.28|
> ||VC|0.83|0.86|0.21|
> |w/o reference|IF|0.82|0.84|0.30|
> ||VC|0.84|0.86|0.21|
>
> As shown above, removing the reference images leads to only negligible changes, while the Oracle still maintains high agreement with human judgments. This indicates that the reported performance is **not materially driven by access to the Gemini-generated golden answers**.
>
> >W3: Consideration for the diversity of answers.
>
> A: We agree with the reviewer that image editing is inherently open-ended and that multiple valid outputs may exist for the same instruction. However, in our Oracle mode, **the edited image is not judged by visual similarity to a single reference image**. As clarified in W2, the evaluation is based on our predefined instruction-following criteria, including successful localization, correct editing action, and preservation of non-target regions. Therefore, visually diverse edits can all receive high scores as long as they satisfy the instruction.
>
> Moreover, our reference-free ablation in W2 suggests that the Oracle is unlikely to misjudge high-quality but visually different results.
>
> >W4: Theoretical analysis of model performance and future improvement suggestions.
>
> A: We agree that understanding the fundamental reasons behind these performance drops is a valuable research direction.
>
> However, we would like to clarify the scope of this work. Similar to other benchmark papers [1,2], our contributions lie in providing **high-quality test samples, establishing a reliable evaluation protocol, and empirically exposing the current shortcomings of state-of-the-art models.**
>
> **A rigorous theoretical analysis of the underlying reasons for performance differences across models would require substantial model-specific investigation of architectures, representations, and training pipelines.** While such analysis would be valuable, it goes beyond the intended scope of this benchmark paper.
>
> [1] ImgEdit: A Unified Image Editing Dataset and Benchmark.
>
> [2] MagicBrush: A Manually Annotated Dataset for Instruction-Guided Image Editing.

---

> > ### Author Rebuttal · Reviewer_kEx7 · 2026-04-03
> >
> > My concerns have been fully addressed. I will raise my score.

---

> > > ### Author Response · Authors · 2026-04-05
> > >
> > > Dear Reviewer kEx7,
> > >
> > > We sincerely thank you for your positive feedback and for raising the score. We are glad that our responses have fully addressed your concerns. We greatly appreciate your time and careful review of our paper.
> > >
> > > Best regards,
> > >
> > > The Authors

---

### Official Review · Reviewer_B8tf · 2026-03-12

**Soundness:** 3
**Presentation:** 4
**Significance:** 3
**Originality:** 3
**Overall Recommendation:** 5
**Confidence:** 4

**Summary:**

The paper identifies that existing benchmarks for image editing are biased toward salient, large-scale objects. To bridge this gap, the authors construct DLEBench: A dataset featuring seven instruction types where targets are minute. develop a 3-Stage Pipeline: Metadata construction via LMMs, reference image generation using a specialized cropping mechanism, and rigorous human verification. Propose a New Evaluation Protocol: Moving away from vague LMM-as-a-judge rubrics toward a hierarchical, failure-mode-based scoring system.

**Compliance With Llm Reviewing Policy:**

Affirmed.

**Final Justification:**

The authors’ rebuttal addressed most of my concerns. I therefore maintain my overall positive recommendation of accept.

**Key Questions For Authors:**

- When using the crop-and-edit strategy to generate reference images, did you observe seam artifacts or lighting inconsistencies at crop boundaries? If so, could these artifacts influence the Visual Consistency scores of the evaluated models?

- Most evaluated models are tested in a zero-shot setting. Do you have any insights into whether fine-tuning on DLEBench-style data could improve small-object editing, or whether this limitation mainly stems from model resolution or architectural constraints?

Also see the weakness section.

**Limitations:**

yes. The authors adequately discussed the limitations.

**Strengths And Weaknesses:**

Strengths

- Focusing on small-scale objects is a practical and technically challenging direction. The proposed benchmark and evaluation protocol introduce a novel task and are likely to be useful for the community.

- The “crop-and-edit” strategy for generating references is a reasonable workaround for the localization failures commonly observed in current diffusion models.

- The dual-mode evaluation (Tool-driven vs. Oracle-guided) makes sense. It acknowledges that LMM-based judges can have “vision” limitations and provides a way to mitigate them using external tools such as GroundingDINO and Zoom-In.

- The paper includes comprehensive benchmarking across both the evaluation models (judges) and the diffusion models being tested.


Weaknesses

- It would be helpful to analyze the minimum object scale at which models can reliably perform edits. For example, selecting one or two representative models and plotting a performance–object size curve would provide insight into how editing capability degrades as objects become smaller.

---

> ### Author Rebuttal · Authors · 2026-03-31
>
> Thanks for your valuable review, feedback, and recognition of our work. This has given us great encouragement.
>
> >W1: Suggestion to analyze the minimum object scale for reliable edits by plotting a performance-object size curve.
>
> A: We thank the reviewer for this helpful suggestion. We have conducted such analysis in **Figure 6 of the paper** (Line 330). Due to page constraints, rather than providing a separate standalone plot, we included a curve of performance versus object size as an inset in the top-left corner of each model’s sub-figure.
>
> As observed in these curves, we have the following insights:
>
> - **Performance Bottlenecks**: Models such as MagicBrush and UniWorld-V1 exhibit consistently poor performance regardless of the object size. Their performance curves remain relatively flat and low, suggesting that the small-scale object editing tasks in DLEBench exceed their current capabilities even at the larger end of our defined small-scale spectrum.
>
> - **Scale-Dependent Performance**: In contrast, for the remaining models, there is a clear positive correlation between object size and performance. As the target object becomes larger, the scores generally increase. Notably, when the edited object occupies more than 1% of the image area, Gemini-3-Pro begins to demonstrate relatively reliable editing performance, whereas the other models still struggle to achieve reliable results.
>
> >Q1: Question regarding boundary artifacts caused by crop-and-edit and their impact on visual consistency scores.
>
> A: We thank the reviewer for raising this important question. We indeed observed that the crop-and-edit strategy can introduce seam artifacts near crop boundaries (as also noted in Line 195 of the paper), which can negatively affect visual consistency.
>
> To mitigate this issue, we adopted a two-fold strategy. **First**, we introduced an adaptive bbox expansion method (Line 200) to reduce the likelihood of boundary artifacts. **Second**, we incorporated a Human Verification stage into the data construction pipeline. During the Visual Verification step (Line 186), annotators inspect generated images for seam artifacts. If such issues are detected, the image is regenerated up to three times; if the problem persists, the sample is discarded.
>
> Therefore, although such artifacts may arise during intermediate generation, they are largely filtered out before inclusion in DLEBench.
>
> >Q2: Question about whether the limitation stems from a lack of fine-tuning or fundamental resolution/architectural constraints.
>
> A: We thank the reviewer for this insightful question. While task-specific fine-tuning may improve performance to some extent, **we believe that the more fundamental limitation lies in model resolution and architectural constraints.**
>
> Current state-of-the-art IIEMs are predominantly built on two architectural paradigms, both of which face intrinsic difficulties when handling very small objects:
>
> - **Diffusion models**: Most diffusion-based editors operate in a compressed latent space. For very small targets (e.g., around 1% of the image area), the corresponding latent representation can become extremely limited. This makes precise localization difficult and may discard crucial structural and semantic details, often leading to over-modification.
>
> - **Autoregressive models**: These models typically rely on visual tokenizers that partition images into discrete patches. Very small objects may therefore be represented by only a few visual tokens, which severely limits accurate localization and fine-grained manipulation.
>
> In summary, although fine-tuning on DLEBench-style data may bring some improvement, we argue that substantially improving small-object editing will likely require architectural advances, such as **decoupling visual perception from image manipulation or dynamically allocating higher token density and attention to target regions**.

---

> > ### Author Rebuttal · Reviewer_B8tf · 2026-04-04
> >
> > I appreciate the author’s rebuttal. It addressed most of my concerns. Therefore, I will keep my current positive rating.

---

> > > ### Author Response · Authors · 2026-04-05
> > >
> > > Dear Reviewer B8tf,
> > >
> > > Thank you very much for your positive feedback and appreciation of our work. We are greatly encouraged by your comments. We sincerely appreciate your time and careful review of our paper.
> > >
> > > Best regards,
> > >
> > > The Authors

---

### Decision · Program_Chairs · 2026-04-30

**Decision:**

Accept (regular)

**Comment:**

This paper receives final ratings of (5, 4, 4, 4). Reviewers kEx7 and GhMP increased their scores after the rebuttal. The reviewers generally agreed that the authors’ additional experiments successfully demonstrated the framework's robustness. The reviewers generally agree that DLEBench provides a novel, technically solid, and valuable benchmark for the under-explored area of small-scale object editing. The AC finds no reason to overturn the decision of the reviewers.